# Federated Learning with Bilateral Curation for Partially Class-Disjoint Data

**Ziqing Fan[1,2], Ruipeng Zhang[1,2], Jiangchao Yao[1,2], Bo Han[3], Ya Zhang[1,2], Yanfeng Wang[1,2,✉]**
[1]Cooperative Medianet Innovation Center, Shanghai Jiao Tong University,
[2]Shanghai AI Laboratory, [3]Hong Kong Baptist University
{zqfan_knight, zhangrp, sunarker}@sjtu.edu.cn
bhanml@comp.hkbu.edu.hk, {ya_zhang, wangyanfeng}@sjtu.edu.cn

## Abstract

Partially class-disjoint data (PCDD), a common yet under-explored data formation where each client contributes *a part of classes* (instead of all classes) of samples, severely challenges the performance of federated algorithms. Without full classes, the local objective will contradict the global objective, yielding the angle collapse problem for locally missing classes and the space waste problem for locally existing classes. As far as we know, none of the existing methods can intrinsically mitigate PCDD challenges to achieve holistic improvement in the bilateral views (both global view and local view) of federated learning. To address this dilemma, we are inspired by the strong generalization of simplex Equiangular Tight Frame (ETF) on the imbalanced data, and propose a novel approach called FedGELA where the classifier is globally fixed as a simplex ETF while locally adapted to the personal distributions. Globally, FedGELA provides fair and equal discrimination for all classes and avoids inaccurate updates of the classifier, while locally it utilizes the space of locally missing classes for locally existing classes. We conduct extensive experiments on a range of datasets to demonstrate that our FedGELA achieves promising performance (averaged improvement of 3.9% to FedAvg and 1.5% to best baselines) and provide both local and global convergence guarantees. Source code is available at: https://github.com/MediaBrain-SJTU/FedGELA.

## 1 Introduction

Partially class-disjoint data (PCDD) [13, 18, 21] refers to an emerging situation in federated learning [14, 22, 43, 46, 50] where each client only possesses information on a subset of categories, but all clients in the federation provide the information on the whole categories. For instance, in landmark detection [39] for thousands of categories with data locally preserved, most contributors only have a *subset* of categories of landmark photos where they live or traveled before; and in the diagnosis of Thyroid diseases, due to regional diversity different hospitals may have shared and distinct Thyroid diseases [10]. It is usually difficult for each party to acquire the full classes of samples, as the participants may be lack of domain expertise or limited by demographic discrepancy. Therefore, how to efficiently handle the *partially class-disjoint data* is a critical (yet under-explored) problem in real-world federated learning applications for the pursuit of personal and generic interests.

Prevalent studies mainly focus on the general heterogeneity without specially considering the PCDD challenges: generic federated leaning (G-FL) algorithms adopt a uniform treatment of all classes and mitigate personal differences by imposing constraints on local training [17, 19], modifying logits [21, 47] adjusting the weights of submitted gradients [37] or generating synthetic data [54]; in contrast, personalized federated learning (P-FL) algorithms place relatively less emphasis on locally missing classes and selectively share either partial network parameters [1, 6] or class prototypes [33] to minimize the impact of personal characteristics, thereby separating the two topics. Those methods

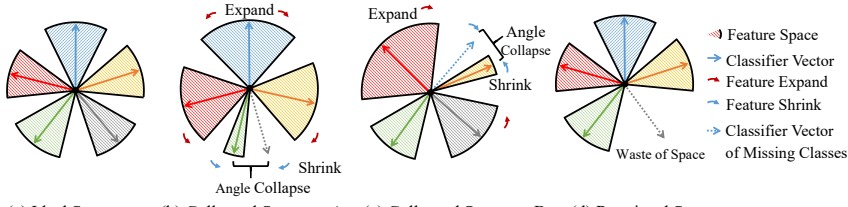

(a) Ideal Structure    (b) Collapsed Structure A    (c) Collapsed Structure B    (d) Restricted Structure

Figure 1: Illustration of feature spaces and classifier vectors trained on the global dataset, two partially class-disjoint datasets (A and B), and restricted by federated algorithms. (a) is trained on the globally balanced dataset with full classes. (b) and (c) are trained on datasets A and B, respectively, which suffer from different patterns of classifier angle collapse problems. (d) is averaged in the server or constrained by some federated algorithms.

might directly or indirectly help mitigate the data shifts caused by PCDD, however, as far as we know, none of the existing works can mitigate the PCDD challenges to achieve holistic improvement in the bilateral views (global and local views) of federated learning. Please refer to Table 1 for a comprehensive comparison among a range of FL methods from different aspects.

Without full classes, the local objective will contradict the global objective, yielding the angle collapse for locally missing classes and the waste of space for locally existing classes. Ideally, as shown in Figure 1(a), global features and their corresponding classifier vectors shall maintain a proper structure to pursue the best separation of all classes. However, the angles of locally missing classes' classifier vectors will collapse, when trained on each client with partially class-disjoint data, as depicted in Figure 1(b), 1(c). FedRS [21] notices the degenerated updates of the classifier and pursues the same symmetrical structure in the local by restricting logits of missing classes. Other traditional G-FL algorithms indirectly restrict the classifier by constraining logits, features, or model weights, which may also make effects on PCDD. However, they cause another problem: space waste for personal tasks. As shown in Figure 1(d), restricting local structure will waste feature space and limit the training of the local model on existing classes. P-FL algorithms utilize the wasted space by selectively sharing part of models but exacerbate the angle collapse of classifier vectors. Recent FedRod [3] attempts to bridge the gap between P-FL and G-FL by introducing a two-head framework with logit adjustment in the G-head, but still cannot address the angle collapse caused by PCDD.

To tackle the PCDD dilemma from both P-FL and G-FL perspectives, we are inspired by a promising classifier structure, namely *simplex equiangular tight frame* (ETF) [9, 26, 41], which provides each class the same classification angle and generalizes well on imbalanced data. Motivated by its merits, we propose a novel approach, called **FedGELA**, in which the classifier is **Globally** fixed as a simplex **ETF** while **Locally Adapted** to personal tasks. In the global view, FedGELA merges class features and their corresponding classifier vectors, which converge to ETF. In the local view, it provides existing major classes with larger feature spaces and encourages to utilize the spaces wasted by locally missing classes. With such a bilateral curation, we can explicitly alleviate the impact caused by PCDD. In a nutshell, our contributions can be summarized as the following three points:

- We study a practical yet under-explored data formation in real-world applications of federated learning, termed as partially class-disjoint data (PCDD), and identify the angle collapse and space waste challenges that cannot be efficiently solved by existing prevalent methods (Sec. 3.2).
- We propose a novel method called FedGELA that classifier is globally fixed as a symmetrical structure ETF while locally adapted by personal distribution (Sec. 3.3), and theoretically show the local and global convergence analysis for PCDD with the experimental verification (Sec. 4.2).
- We conduct a range of experiments on multiple benchmark datasets under the PCDD case and a real-world dataset to demonstrate the bilateral advantages of FedGELA over the state-of-the-art methods from multiple views like the larger scale of clients and straggler situations (Sec. 5.2). We also provide further analysis like classification angles during training and ablation study. (Sec. 5.3).

## 2 Related Work

### 2.1 Partially Class-Disjoint Data and Federated Learning algorithms

Partially class-disjoint data is one common formation among clients that can significantly impede the convergence, performance, and efficiency of algorithms in FL [18]. It belongs to the data heterogeneity

Table 1: Key differences between SOTA methods and our FedGELA categorized by targets (P-FL or G-FL), techniques (improve from the views of features, logits or model), and whether directly mitigate angle collapse of classifier vectors or save locally wasted feature spaces caused by PCDD.

| Target | Research work | Feature View | Logit View | Model View | Mitigate Collapse | Save Space |
|--------|---------------|--------------|------------|------------|-------------------|------------|
| G-FL | FedProx | - | - | ✓ | ✓ | - |
| | MOON | ✓ | - | - | - | - |
| | FedRS | - | ✓ | - | ✓ | - |
| | FedGen | ✓ | - | - | ✓ | - |
| | FedLC | - | ✓ | - | - | - |
| P-FL | FedRep | ✓ | - | ✓ | - | ✓ |
| | FedProto | ✓ | - | ✓ | - | ✓ |
| | FedBABU | ✓ | - | ✓ | - | ✓ |
| G&P-FL | FedRod | - | ✓ | ✓ | - | ✓ |
| | FedGELA(ours) | ✓ | ✓ | ✓ | ✓ | ✓ |

case, but does have a very unique characteristic different from the ordinary heterogeneity problem. That is, if only each client only has a subset of classes, it does not share the optimal Bayes classifier with the global model that considers all classes on the server side. Recently, FedRS [21] has recognized the PCDD dilemma and directly mitigate the angle collapse issue by constraining the logits of missing classes. FedProx [19] also can lessen the collapse by constraining local model weights to stay close to the global model. Other G-FL algorithms try to address data heterogeneity from a distinct perspective. MOON [17] and FedGen [54] utilizes contrastive learning and generative learning to restrict local representations. And FedLC [47] introduces logit calibration to adjust the logits of the local model to match those of the global model, which might indirectly alleviate the angle collapse in the local. However, they all try to restrict local structure as global, resulting in the waste space for personal tasks shown in Figure 1(d). P-FL algorithms try to utilize the wasted space by encouraging the angle collapse of the local classifier. FedRep [6] only shares feature extractors among clients and FedProto [33] only submits class prototypes to save communication costs and align the feature spaces. In FedBABU [25], the classifier is randomly initialized and fixed during federated training while fine-tuned for personalization during the evaluation. However, they all sacrifice the generic performance on all classes. FedRod [3] attempts to bridge this gap by introducing a framework with two heads and employing logit adjustment in the global head to estimate generic distribution but cannot address angle collapse. In Table 1, we categorize these methods by targets (P-FL or G-FL), skews (feature, logit, or model weight), and whether they directly mitigate the angle collapse of local classifier or saving personal spaces for personal spaces. It is evident that none of these methods, except ours, can handle the PCDD problem in both P-FL and G-FL. Furthermore, FedGELA is the only method that can directly achieve improvements from all views.

## 2.2 Simplex Equiangular Tight Frame

The simplex equiangular tight frame (ETF) is a phenomenon observed in neural collapse [26], which occurs in the terminal phase of a well-trained model on a balanced dataset. It is shown that the last-layer features of the model converge to within-class means, and all within-class means and their corresponding classifier vectors converge to a symmetrical structure. To analyze this phenomenon, some studies simplify deep neural networks as last-layer features and classifiers with proper constraints (layer-peeled model) [9, 12, 40, 53] and prove that ETF emerges under the cross-entropy loss. However, when the dataset is imbalanced, the symmetrical structure of ETF will collapse [9]. Some studies try to obtain the symmetrical feature and the classifier structure on the imbalanced datasets by fixing the classifier as ETF [41, 53]. Inspired by this, we propose a novel method called FedGELA that bilaterally curates the classifier to leverage ETF or its variants. See Appendix A for more details about ETF.

## 3 Method

### 3.1 Preliminaries

**ETF under LPM.** A typical L-layer DNN parameterized by $\mathbf{W}$ can be divided into the feature backbone parameterized by $\mathbf{W}^{-L}$ and the classifier parameterized by $\mathbf{W}^{L}$. From the view of layer-peeled model (LPM) [9, 12, 40, 53], training $\mathbf{W}$ with constraints on the weights can be considered

as training the C-class classifier $\mathbf{W}^L = \{\mathbf{W}_1^L, ..., \mathbf{W}_C^L\}$ and features $\mathbf{H} = \{h^1, ..., h^n\}$ of all $n$ samples output by last layer of the backbone with constraints $E_W$ and $E_H$ on them respectively. On the balanced data, any solutions to this model form a simplex equiangular tight frame (ETF) that all last layer features $h_c^{i,*}$ and corresponding classifier $\mathbf{W}_c^{L,*}$ of all classes converge as:

$$\frac{h_c^{i,*}}{\sqrt{E_H}} = \frac{\mathbf{W}_c^{L,*}}{\sqrt{E_W}} = m_c^*, \tag{1}$$

where $m_c^*$ forms the ETF defined as $\mathbf{M} = \sqrt{\frac{C}{C-1}} \mathbf{U} \left( \mathbf{I}_C - \frac{1}{C} \mathbf{1}_C \mathbf{1}_C^T \right)$. Here $\mathbf{M} = [m_1^*, \cdots, m_C^*] \in \mathbb{R}^{d \times C}, \mathbf{U} \in \mathbb{R}^{d \times C}$ allows a rotation and satisfies $\mathbf{U}^T \mathbf{U} = \mathbf{I}_C$ and $\mathbf{1}_C$ is an all-ones vector. ETF is an optimal classifier and feature structure in the balanced case of LPM.

**FedAvg.** On the view of LPM, given N clients and each with $n_k$ samples, the vanilla federated learning via FedAvg consists of four steps [22]: 1) In round $t$, the server broadcasts the global model $\mathbf{W}^t = \{\mathbf{H}^t, \mathbf{W}^{L,t}\}$ to clients that participate in the training (Note that here $\mathbf{H}$ is actually the global backbone $\mathbf{W}^{-L,t}$ instead of real features); 2) Each local client receives the model and trains it on the personal dataset. After $E$ epochs, we acquire a new local model $\mathbf{W}_k^t$; 3) The updated models are collected to the server as $\{\mathbf{W}_1^t, \mathbf{W}_2^t, \ldots, \mathbf{W}_N^t\}$; 4) The server averages local models to acquire a new global model as $\mathbf{W}^{t+1} = \sum_{k=1}^N p_k \mathbf{W}_k^t$, where $p_k = n_k / \sum_{k'=1}^N n_{k'}$. When the pre-defined maximal round $T$ reaches, we will have the final optimized global model $\mathbf{W}^T$.

### 3.2 Contradiction and Motivation

**Contradiction.** In G-FL, the ideal global objective under LPM of federated learning is described as:

$$\min_{\mathbf{H}, \mathbf{W}^L} \sum_{k=1}^N p_k \frac{1}{n_k} \sum_{c \in C_k} \sum_{i=1}^{n_{k,c}} \mathcal{L}_{CE}\left(h_{k,c}^i, \mathbf{W}^L\right).$$

Assuming global distribution is balanced among classes, no matter whether local datasets have full or partial classes, the global objective with constraints on weights can be simplified as:

$$\min_{\mathbf{H}, \mathbf{W}^L} \frac{1}{n} \sum_{c=1}^C \sum_{i=1}^{n_c} \mathcal{L}_{CE}\left(h_c^i, \mathbf{W}^L\right), \text{ s.t. } \left\|\mathbf{W}_c^L\right\|^2 \leqslant E_W, \left\|h_c^i\right\|^2 \leqslant E_H. \tag{2}$$

Similarly, the local objective of k-th client with a set of classes $C_k$ can be described as:

$$\min_{\mathbf{H}_k, \mathbf{W}_k^L} \frac{1}{n_k} \sum_{c \in C_k} \sum_{i=1}^{n_{k,c}} \mathcal{L}_{CE}\left(h_{k,c}^i, \mathbf{W}_k^L\right), \text{ s.t. } \left\|\mathbf{W}_{k,c}^L\right\|^2 \leqslant E_W, \left\|h_{k,c}^i\right\|^2 \leqslant E_H. \tag{3}$$

When PCDD exists ($C_k \neq C$), we can see the contradiction between local and global objectives, which respectively forms two structures, shown in Figure 3(a) and Figure 3(b). After aggregated in server or constrained by some FL methods, the structure in the local is restricted to meet the global structure, causing space waste for personal tasks shown in Figure 1(d).

**Motivation.** To verify the contradiction and related feature and classifier structures, we split CIFAR10 into 10 clients and perform FedAvg on it with Dirichlet Distribution (Dir ($\beta = 0.1$)). As illustrated in Figure 2, the angle difference between existing classes and between missing classes becomes smaller and converges to a similar value in the global model. However, in the local training, angles between existing classes become larger while angles between missing classes become smaller, which indicates the contradiction. With this observation, to bridge the gap between Eq (3) and Eq (2) under PCDD, we need to construct the symmetrical and uniform classifier angles for all classes while encouraging local clients to expand existing classes' feature space. Therefore, we propose our method **FedGELA** that classifier can be **Globally** fixed as **ETF** but **Locally Adapted** based on the local distribution matrix to utilize the wasted space for the existing classes.

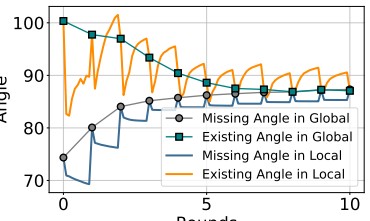

Figure 2: Averaged angles of classifier vectors between locally existing classes (existing angle) and between locally missing classes (missing angle) on CIFAR10 (Dir ($\beta = 0.1$)) in local client and aggregated in global server (local epoch is 10). In global, "existing" angle and "missing" angle converge to similar values while in the local, "existing" angle expands but "missing" angle shrinks.

### 3.3 FedGELA

**Global ETF.**  Given the global aim of achieving an unbiased classifier that treats all classes equally and provides them with the same discrimination and classifier angles, we curate the global model's classifier as a randomly initialized simplex ETF with scaling $\sqrt{E_w}$ at the start of federated training:

$$\mathbf{W}^L = \sqrt{E_w}\mathbf{M}.$$

Then the ETF is distributed to all clients to replace their local classifiers. In Theorem 1, we prove in federated training under some basic assumptions, by fixing the classifier as a randomly simplex ETF with scaling $\sqrt{E_W}$ and constraints $E_H$ on the last layer features, features output by last layer of backbone and their within class means will converge to the ETF similar to Eq (1), which meets the requirement of global tasks.

**Local Adaptation.**  However, when PCDD exists in the local clients, naively combining ETF with FL does not meet the requirement of P-FL as analyzed in Eq (2) and Eq (3). To utilize the wasted space for locally missing classes, in the training stage, we curate the length of ETF received from the server based on the local distribution as below:

$$\mathbf{W}_k^L = \mathbf{\Phi}_k \mathbf{W}^L = \mathbf{\Phi}_k \sqrt{E_w}\mathbf{M}, \tag{4}$$

where $\mathbf{\Phi}_k$ is the distribution matrix of k-th client. Regarding the selection of $\mathbf{\Phi}_k$, it should satisfy a basic rule for federated learning, wherein the aggregation of local classifiers aligns with the global classifier, thereby ensuring the validity of theoretical analyses from both global and local perspectives. Moreover, it is highly preferable for the selection process to avoid introducing any additional privacy leakage risks. To meet the requirement that averaged classifier should be standard ETF: $\mathbf{W}^L = \sum_{k=1}^{N} p_k \mathbf{W}_k^L$ in the globally balanced case, its row vectors are all one's vector multiple statistical values of personal distribution:$(\mathbf{\Phi}_k^T)_c = \frac{n_{k,c}}{n_k \gamma}\mathbf{1}$ ($\gamma$ is a constant, and $n_{k,c}$ and $n_k$ are the c-th class sample number and total sample number of the k-th client) respectively. We set $\gamma$ to $\frac{1}{|C|}$. Finally, the local objective from Eq. (3) is adapted as:

$$\min_{\mathbf{H}_k} \quad \frac{1}{n_k}\sum_{c=1}^{C}\sum_{i=1}^{n_{k,c}} -\log \frac{\exp(\mathbf{\Phi}_{k,c}\mathbf{W}_c^{L\,T} h_{k,c}^i)}{\sum_{c'\in C_k}\exp(\mathbf{\Phi}_{k,c'}\mathbf{W}_{c'}^{L\,T} h_{k,c}^i)},$$
$$\text{s.t.} \quad \|h^i\|^2 \leqslant E_H, \forall 1 \leqslant i \leqslant n_k. \tag{5}$$

**Total Framework.**  After introducing two key parts of FedGELA (Global ETF and Local Adaptation), we describe the total framework of FedGELA. As illustrated and highlighted in Algorithm 1 (refer to Appendix D for the workflow figure), at the initializing stage, the server randomly generates an ETF as the global classifier and sends it to all clients while local clients adjust it based on the personal distribution matrix as Eq (4). At the training stage, local clients receive global backbones and train with adapted ETF in parallel. After $E$ epochs, all clients submit personal backbones to the server. In the server, personal backbones are received and aggregated to a generic backbone, which is broadcast to all clients participating in the next round. At the inference stage, on the client side, we obtain a generic backbone with standard ETF to handle the world data while on the client side, a personal backbone with adapted ETF to handle the personal data.

---

**Algorithm 1** FedGELA

**Input**:$(N, K, n_k, c_k, \mathbf{H}^0, \mathbf{M}, E_W, E_H, T, \eta, E)$

**Parallely for all clients:**  $\mathbf{W}_k^L \leftarrow \mathbf{\Phi}_k \sqrt{E_W}\mathbf{M}$.

**for** $t = 0, 1, \ldots, T-1$ **do**

  ▷ on the server side

  $\mathbf{H}^t \leftarrow \sum_{k=1}^{K} p_k^t H_k^{t-1}$.

  sample K clients from all N clients.

  ▷ on the client side

  **do in parallel for** $\forall k \in K$ **clients**

    receive $\mathbf{H}^t$ from server, $\mathbf{H}_k^t \leftarrow \mathbf{H}^t$.

    **for** $\tau = 0, 1, ..., E-1$ **do**

      sample a mini-batch $b_k^{tE+\tau}$ in local data.

      $H_k^t \leftarrow H_k^t - \eta\nabla F_k(b_k^t, \mathbf{\Phi}_k W_g^L; \mathbf{H}_k^t)$

    **end for**

    submit $\mathbf{H}_k^t$ to the server.

  **end in parallel**

**end for**

**Output**:$(\mathbf{H}^T, W_g^L)$ and $(\mathbf{H}_k^T, \mathbf{\Phi}_k W_g^L)$.

---

## 4  Theoretical Analysis

In this part, we first primarily introduce some notations and basic assumptions in Sec. 4.1 and then present the convergence guarantees of both local models and the global model under the PCDD with the proper empirical justification and discussion in Sec. 4.2. (Please refer to Appendix B for entire proofs and Appendix D for details on justification experiments.)

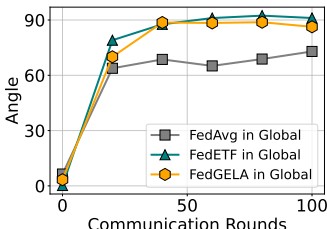 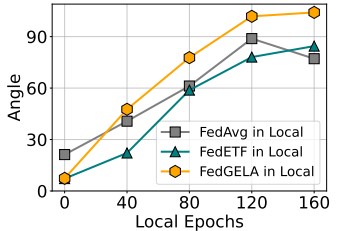 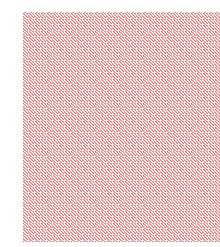

(a) Global convergence verification.    (b) Local convergence verification.    (c) Adapted structure.

Figure 3: Illustration of local and global convergence verification together with the effect of $\mathbf{\Phi}$. (a) and (b) are the results of averaged angle between all class means and between locally existing class means in FedAvg, FedGE, and FedGELA on CIFAR10 under 50 clients and Dir ($\beta = 0.2$). (c) is the illustration of how local adaptation utilizes the wasted space of missing classes for existing classes.

### 4.1 Notations

We use $t$ and $T$ to denote a curtain round and pre-defined maximum round after aggregation in federated training, $tE$ to denote the state that just finishing local training before aggregation in round $t$, and $tE + \tau$ to denote $\tau$-th local iteration in round $t$ and $0 \leq \tau \leq E - 1$. The convergence follows some common assumptions in previous FL studies and helpful math results [15, 20, 29–31, 33, 36, 38, 45, 51] including smoothness, convexity on loss function $F_1, F_2, \cdots, F_N$ of all clients, bounded norm and variance of stochastic gradients on their gradient functions $\nabla F_1, \nabla F_2, \cdots, \nabla F_N$ and heterogeneity $\Gamma_1$ reflected as the distance between local optimum $\mathbf{W}_k^*$ and global optimum $\mathbf{W}^*$. Please refer to the concrete descriptions of those assumptions in Appendix B. Besides, in Appendix B, we additionally provide a convergence guarantee without a bounded norm of stochastic gradients, as some existing works [24, 32] point out the contradiction to the strongly convex.

### 4.2 Convergence analysis

Here we provide the global and local convergence guarantee of our FedGELA compared with FedAvg and FedGE (FedAvg with only the Globally Fixed ETF) in Theorem 1 and Theorem 2. To better explain the effectiveness of our FedGELA in local and global tasks, we record the averaged angle between all class means in global and existing class means in local as shown in Figure 3(a) and Figure 3(b). Please refer to Appendix B for details on the proof and justification of theorems.

**Theorem 1** (Global Convergence). *If $F_1, ..., F_N$ are all L-smooth, $\mu$-strongly convex, and the variance and norm of $\nabla F_1, ..., \nabla F_N$ are bounded by $\sigma$ and $G$. Choose $\kappa = L/\mu$ and $\gamma = \max\{8\kappa, E\}$, for all classes $c$ and sample $i$, expected global representation by cross-entropy loss will converge to:*

$$\mathbb{E}\left[\log \frac{(\mathbf{W}^{L,*})^T h_c^{i,*}}{(\mathbf{W}_g^L)^T h_c^i}\right] \leq \frac{\kappa}{\gamma + T - 1}\left(\frac{2B}{\mu} + \frac{\mu\gamma}{2}\mathbb{E}||\mathbf{W}^1 - \mathbf{W}^*||^2\right),$$

*where in FedGELA, $B = \sum_{k=1}^{N}(p_k^2\sigma^2 + p_k||\mathbf{\Phi}_k\mathbf{W}^L - \mathbf{W}^L||) + 6L\Gamma_1 + 8(E-1)^2G^2$. Since $\mathbf{W}^L = \mathbf{W}^{L,*}$ and $(\mathbf{W}^{L,*})^T h_{c_i}^{i,*} \geq \mathbb{E}[(\mathbf{W}^L)^T h_{c_i}^i]$, $h_{c_i}^i$ will converge to $h_{c_i}^{i,*}$.*

In Theorem 1, the variable $B$ represents the impact of algorithmic convergence ($p_k^2\sigma^2$), non-iid data distribution ($6L\Gamma_1$), and stochastic optimization ($8(E-1)^2G^2$). The only difference between FedAvg, FedGE, and our FedGELA lies in the value of $B$ while others are kept the same. FedGE and FedGELA have a smaller $G$ compared to FedAvg because they employ a fixed ETF classifier that is predefined as optimal. FedGELA introduces a minor additional overhead ($p_k||\mathbf{\Phi}_k\mathbf{W}^L - \mathbf{W}^L||$) on the global convergence of FedGE due to the incorporation of local adaptation to ETFs. The cost might be negligible, as $\sigma$, $G$, and $\Gamma_1$ are defined on the whole model weights while $p_k||\mathbf{\Phi}_k\mathbf{W}^L - \mathbf{W}^L||$ is defined on the classifier. To verify this, we conduct experiments in Figure 3(a), and as can be seen, FedGE and FedGELA have similar quicker speeds and larger classification angles than FedAvg.

**Theorem 2** (Local Convergence). *If $F_1, ..., F_N$ are L-smooth, variance and norm of their gradients are bounded by $\sigma$ and $G$, and the heterogeneity is bounded by $\Gamma_1$, clients' expected local loss satisfies:*

$$\mathbb{E}[F_k^{(t+1)E}] \leq F_k^{tE} + \frac{LE\eta_t^2}{2}\sigma^2 + \Gamma_1 - A,$$

*where in FedGELA, $A = (\eta_t - \frac{L}{2}\eta_t^2)EG^2 - L\left\|\mathbf{\Phi}_k\mathbf{W}^L - \mathbf{W}^L\right\|$, which means if $A - \frac{G^4}{LE(G^2+\sigma^2)} \leq 0$, there exist learning rate $\eta_t$ making the expected local loss decreasing and converging.*

In Theorem 2, only "A" is different on the convergence among FedAvg, FedGE, and FedGELA. Fixing the classifier as ETF and adapting the local classifier will introduce smaller G and additional cost of $L \left\| \mathbf{\Phi}_k \mathbf{W}^L - \mathbf{W}^L \right\|$ respectively, which might limit the speed of local convergence. However, FedGELA might reach better local optimal by adapting the feature structure. As illustrated in Figure 3 (c), the adapted structure expands the decision boundaries of existing major classes and better utilizes the feature space wasted by missing classes. To verify this, in Figure 3(b), we record the averaged angles between the existing class means during the local training. It can be seen that FedGELA converges to a much larger angle than both FedAvg and FedGE, which suits our expectations. More angle results can be seen in Figure 5.

## 5 Experiments

### 5.1 Experimental Setup

**Datasets.** We adopt three popular benchmark datasets SVHN [23], CIFAR10/100 [16] in federated learning. As for data splitting, we utilize Dirichlet Distribution (Dir ($\beta$), $\beta = \{10000, 0.5, 0.2, 0.1\}$) to simulate the situations of independently identical distribution and different levels of PCDD. Besides, one standard real-world PCDD dataset, Fed-ISIC2019 [4, 7, 34, 35] is used, and we follow the setting in the Flamby benchmark [34]. Please refer to Appendix C for more details.

**Metrics.** Denote PA as the personal accuracy, which is the mean of the accuracy computed on each client test dataset, and GA as the generic accuracy on global test dataset (mixed clients' test datasets). Since there is no global model in P-FL methods, we calculate GA of them as the averaged accuracy of all best local models on global test dataset, which is the same as FedRod [3]. Regarding PA, we record the best results of personal models for P-FL methods while for G-FL methods we fine-tune the best global model in 10 epochs and record the averaged accuracy on all client test datasets. For FedRod and FedGELA, we can directly record the GA and PA (without fine-tuning) during training.

**Implementation.** We compare FedGELA with FedAvg, FedRod [3], multiple state-of-the-art methods in G-FL (FedRS [21], MOON [17], FedProx [19], FedGen [54] and FedLC [47]) and in P-FL (FedRep [6], FedProto [33] and FedBABU [25]). For SVHN, CIFAR10, and CIFAR100, we adopt a commonly used ResNet18 [8, 17, 47, 48, 52] with one FC layer as the backbone, followed by a layer of classifier. FedGELA replaces the classifier as a simple ETF. We use SGD with learning rate 0.01, weight decay $10^{-4}$, and momentum 0.9. The batch size is set as 100 and the local updates are set as 10 epochs for all approaches. As for method-specific hyper-parameters like the proximal term in FedProx, we tune it carefully. In our method, there are $E_W$ and $E_H$ need to set, we normalize features with length 1 ($E_H = 1$) and only tune the length scaling of classifier ($E_W$). All methods are implemented by PyTorch [27] with NVIDIA GeForce RTX 3090. See detailed information in Appendix C.

### 5.2 Performance of FedGELA

In this part, we compare FedGELA with FedAvg, FedRod, three SOTA methods of P-FL (FedRep, FedProto, and FedBABU), four SOTA methods of G-FL (FedProx, MOON, FedRS, FedLC and FedGen) on different aspects including the scale of clients, the level of PCDD, straggler situations, and real-world applications. Similar to recent studies [8, 17, 44], we split SVHN, CIFAR10, and CIFAR100 into 10 and 50 clients and each round select 10 clients to join the federated training, denoted as full participation and partial participation (straggler situation), respectively. With the help of Dirichlet distribution [11], we verify all methods on IID, Non-IID ($\beta = 0.5$), and extreme Non-IID situations ($\beta = 0.1$ or $\beta = 0.2$). As the decreasing $\beta$, the level of PCDD increases and we show the heat map of data distribution in Appendix C. We set $\beta = 0.2$ in partial participation to make sure each client has at least one batch of samples. The training round for SVHN and CIFAR10 is 50 in full participation and 100 in partial participation while for CIFAR100, it is set to 100 and 200. Besides, we also utilize a real federated scenario Fed-ISIC2019 to verify the ability to real-world application.

**Full participation and partial participation.** As shown in Table 2, with the decreasing $\beta$ or increasing number of clients, the generic performance of FedAvg and all other methods greatly drops while the personal performance of all methods greatly increases. This means under PCDD and the straggler problem, the performance of generic performance is limited but the personal distribution is easier to capture. As for P-FL methods, they fail in global tasks especially in severe PCDD

Table 2: Personal and generic performance on SVHN, CIFAR10, and CIFAR100. We use Dir ($\beta = 0.5$) for medium heterogeneity and Dir ($\beta = 0.1$) or Dir ($\beta = 0.2$) for high-level heterogeneity. To verify the straggler situation, we split all datasets into 10 or 50 clients for full participation or partial participation, and in each round, 10 clients are selected in the federated training.

| Dataset | Method | Full Participation (10, 10) | | | | | | Partial Participation (50, 10) | | | | | |
|---|---|---|---|---|---|---|---|---|---|---|---|---|---|
| | #Partition | IID | | $\beta = 0.5$ | | $\beta = 0.1$ | | IID | | $\beta = 0.5$ | | $\beta = 0.2$ | |
| | #Metric | PA | GA | PA | GA | PA | GA | PA | GA | PA | GA | PA | GA |
| SVHN | FedAvg | 93.01 | 92.61 | 93.95 | 91.24 | 98.10 | 75.24 | 91.44 | 91.29 | 92.70 | 89.29 | 95.31 | 84.70 |
| | FedProx | 93.12 | 93.12 | 93.71 | 92.15 | 97.98 | 75.13 | 91.67 | 91.66 | 92.71 | 89.98 | 95.13 | 85.68 |
| | MOON | 93.16 | 93.16 | 92.98 | 92.46 | 98.06 | 76.21 | 93.49 | 91.41 | 91.86 | 90.20 | 95.78 | 86.22 |
| | FedRS | 93.29 | 93.21 | 93.92 | 92.33 | 98.04 | 76.26 | 91.63 | 91.59 | 93.51 | 91.70 | 96.20 | 87.78 |
| | FedGen | 94.02 | 93.99 | 94.47 | 92.66 | 98.22 | 76.51 | 91.47 | 91.33 | 93.67 | 91.35 | 95.77 | 87.59 |
| | FedLC | 93.29 | 93.28 | 94.76 | 91.20 | 98.24 | 76.17 | 91.69 | 91.67 | 92.73 | 91.02 | 95.20 | 86.92 |
| | FedRep | 93.01 | 92.61 | 94.77 | 91.24 | 97.87 | 68.52 | 91.77 | 89.20 | 93.14 | 80.94 | 95.38 | 67.77 |
| | FedProto | 93.21 | 91.68 | 94.48 | 85.85 | 98.26 | 56.49 | 90.23 | 87.27 | 93.28 | 76.59 | 95.62 | 54.92 |
| | FedBABU | 93.26 | 93.08 | 95.20 | 92.04 | 98.16 | 75.52 | 93.69 | 91.05 | 93.54 | 90.49 | 95.70 | 84.42 |
| | FedRod | 93.50 | 93.22 | 95.47 | 92.09 | 98.06 | 76.24 | 92.04 | 91.65 | 93.96 | 91.20 | 95.68 | 86.98 |
| | **FedGELA** | **94.84** | **94.66** | **96.27** | **93.66** | **98.52** | **78.88** | **94.68** | **93.59** | **95.54** | **93.29** | **96.85** | **89.58** |
| CIFAR10 | FedAvg | 73.17 | 72.8 | 81.67 | 67.28 | 92.66 | 54.57 | 66.88 | 66.64 | 70.64 | 61.81 | 80.04 | 49.13 |
| | FedProx | 73.69 | 73.69 | 81.95 | 67.53 | 92.94 | 56.13 | 67.67 | 67.27 | 73.62 | 60.80 | 80.66 | 50.82 |
| | MOON | 73.29 | 73.29 | 82.27 | 68.34 | 92.90 | 55.61 | 67.58 | 67.58 | 74.64 | 61.81 | 83.42 | 52.19 |
| | FedRS | 73.56 | 72.94 | 81.59 | 68.10 | 92.57 | 58.19 | 66.76 | 66.52 | 72.21 | 58.95 | 81.11 | 51.66 |
| | FedLC | 73.05 | 73.00 | 81.99 | 67.97 | 92.48 | 57.02 | 67.46 | 67.13 | 72.57 | 61.31 | 82.14 | 55.15 |
| | FedGen | 73.72 | 73.49 | 82.22 | 69.33 | 92.79 | 58.04 | 68.74 | 68.02 | 75.52 | 62.44 | 81.07 | 53.46 |
| | FedRep | 73.42 | 73.23 | 83.30 | 47.96 | 92.92 | 38.32 | 67.85 | 67.74 | 77.28 | 42.64 | 84.52 | 33.22 |
| | FedProto | 67.06 | 66.74 | 81.03 | 46.99 | 93.17 | 32.13 | 61.85 | 52.76 | 72.89 | 37.47 | 81.73 | 26.07 |
| | FedBABU | 73.86 | 72.30 | 81.40 | 65.03 | 92.94 | 53.65 | 66.99 | 64.90 | 77.59 | 58.17 | 82.92 | 49.90 |
| | FedRod | 74.24 | 73.76 | 82.34 | 70.74 | 92.27 | 58.86 | 70.09 | 70.04 | 78.23 | 64.13 | 84.63 | 58.86 |
| | **FedGELA** | **75.02** | **74.07** | **84.52** | **72.73** | **94.28** | **61.57** | **72.33** | **72.04** | **80.96** | **65.08** | **86.55** | **60.52** |
| CIFAR100 | FedAvg | 65.27 | 65.27 | 65.59 | 63.96 | 76.43 | 59.17 | 55.16 | 55.29 | 55.36 | 54.15 | 58.85 | 53.39 |
| | FedProx | 65.71 | 65.71 | 65.31 | 64.18 | 75.95 | 59.93 | 56.86 | 56.89 | 56.89 | 56.08 | 59.27 | 55.25 |
| | MOON | 65.33 | 65.33 | 65.23 | 64.79 | 75.45 | 60.12 | 56.91 | 56.86 | 56.72 | 56.14 | 59.51 | 55.53 |
| | FedRS | 65.18 | 65.64 | 66.48 | 64.62 | 76.86 | 60.77 | 56.51 | 56.91 | 56.45 | 56.34 | 61.92 | 55.99 |
| | FedGen | 65.74 | 65.75 | 66.72 | 64.33 | 76.92 | 60.43 | 56.77 | 56.74 | 57.43 | 56.27 | 60.09 | 55.27 |
| | FedLC | 65.83 | 65.84 | 65.91 | 65.02 | 75.67 | 60.07 | 56.87 | 56.04 | 56.56 | 56.28 | 60.89 | 55.95 |
| | FedRep | 61.21 | 59.21 | 67.87 | 52.51 | 77.81 | 42.77 | 53.41 | 51.44 | 55.60 | 48.67 | 67.70 | 33.10 |
| | FedProto | 56.56 | 56.26 | 66.08 | 46.88 | 77.68 | 37.63 | 52.41 | 50.04 | 54.05 | 42.88 | 63.22 | 28.74 |
| | FedBABU | 65.63 | 65.28 | 71.30 | 64.54 | 80.33 | 60.99 | 56.91 | 54.57 | 60.14 | 54.40 | 68.44 | 54.24 |
| | FedRod | 66.17 | 66.17 | 72.05 | 65.19 | 80.46 | 61.01 | 57.76 | 57.01 | 63.90 | 56.53 | 72.37 | 54.67 |
| | **FedGELA** | **67.28** | **68.07** | **72.61** | **66.94** | **82.79** | **63.13** | **61.70** | **59.29** | **64.37** | **58.60** | **72.93** | **58.53** |

Table 3: Personal and generic performance on a real federated application Fed-ISIC2019. More results of other realworld dataset are shown in the Appendix.

| Method | FedAvg | FedProx | MOON | FedRS | FedGen | FedLC | FedRep | FedProto | FedBABU | FedRod | **FedGELA** |
|---|---|---|---|---|---|---|---|---|---|---|---|
| PA | $77.27_{\pm 0.19}$ | $77.91_{\pm 0.16}$ | $77.94_{\pm 0.17}$ | $78.27_{\pm 0.12}$ | $78.02_{\pm 0.23}$ | $77.58_{\pm 0.19}$ | $76.94_{\pm 0.13}$ | $77.80_{\pm 0.17}$ | $\underline{78.91}_{\pm 0.13}$ | $78.65_{\pm 0.34}$ | $\mathbf{79.27}_{\pm 0.19}$ |
| GA | $73.59_{\pm 0.17}$ | $73.69_{\pm 0.26}$ | $73.80_{\pm 0.21}$ | $74.60_{\pm 0.15}$ | $74.37_{\pm 0.27}$ | $74.26_{\pm 0.25}$ | $68.05_{\pm 0.37}$ | $66.26_{\pm 0.16}$ | $74.06_{\pm 0.31}$ | $\underline{74.98}_{\pm 0.21}$ | $\mathbf{75.85}_{\pm 0.16}$ |

situations since they do not consider the global convergence during training. As for G-FL methods, the performance is better in generic tasks but limited in personalized tasks, especially in CIFAR100. They constrain the model's ability to fit personalized distributions during local training, resulting in improved consistency during global optimization. As can be seen, our FedGELA consistently exceeds all baselines for all settings with averaged performance of 2.42%, 5.2% and 5.7% to FedAvg and 1.35%, 1.64% and 1.81% to the best baseline on the three datasets respectively.

**Performance in real-world applications.** Except for the above three benchmarks, we also verify FedGELA with other methods under a real PCDD federated application: Fed-ISIC2019. As shown in Table 3, our method achieves the best improvement of 2% and 2.26% relative to FedAvg and of 0.36% and 1.25% relative to the best baseline on personal and generic tasks respectively, which demonstrates that our method is robust to practical situations in the both views. In the Appendix D.5, we provide more results on additional two real-world applications named FEMNIST and SHAKESPEARE to further show the effectiveness of our method in the real-world scenarios.

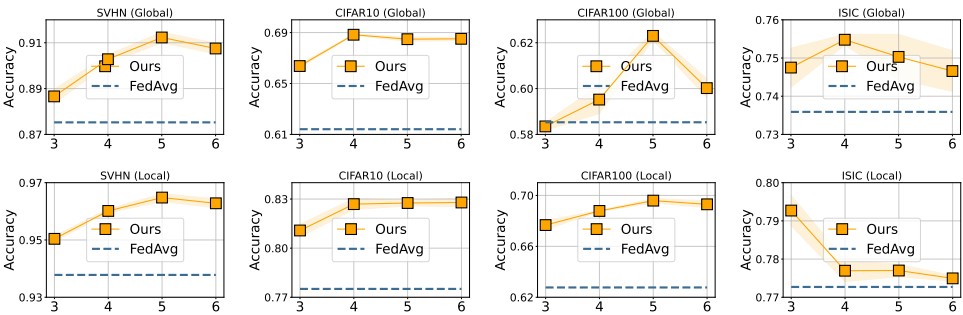

Figure 4: Bilateral performance on four datasets by tuning $logE_W$ (x axis) of FedGELA.

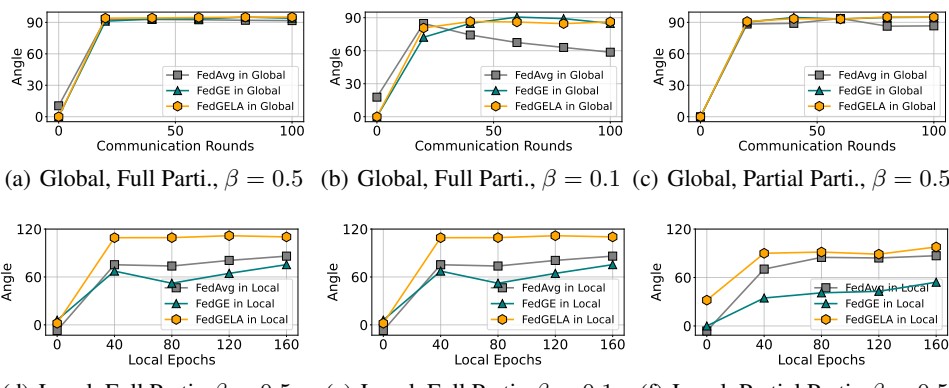

(a) Global, Full Parti., $\beta = 0.5$ (b) Global, Full Parti., $\beta = 0.1$ (c) Global, Partial Parti., $\beta = 0.5$

(d) Local, Full Parti., $\beta = 0.5$ (e) Local, Full Parti., $\beta = 0.1$ (f) Local, Partial Parti., $\beta = 0.5$

Figure 5: Illustration of the averaged angle between locally existing classes and missing classes on the local client and global server of FedAvg, FedGE, and our FedGELA on CIFAR10.

Table 4: Ablation study of FedGELA. GE and LA mean the global ETF and local adaptation.

| GE | LA | SVHN | | | | CIFAR10 | | | | CIFAR100 | | | | Fed-ISIC2019 | |
|---|---|---|---|---|---|---|---|---|---|---|---|---|---|---|---|
| #Partition | | Full Parti. | | Partial Parti. | | Full Parti. | | Partial Parti. | | Full Parti. | | Partial Parti. | | Real World | |
| #Metric | | PA | GA | PA | GA | PA | GA | PA | GA | PA | GA | PA | GA | PA | GA |
| - | - | 95.02 | 86.36 | 93.15 | 88.43 | 82.50 | 64.88 | 72.52 | 59.19 | 69.09 | 62.80 | 56.46 | 54.28 | 77.27 | 73.59 |
| ✓ | - | 95.92 | 88.93 | 93.97 | 92.42 | 83.63 | 69.70 | 77.66 | 65.56 | 71.46 | 66.02 | 62.67 | 58.98 | 69.88 | 75.54 |
| - | ✓ | 95.93 | 74.84 | 93.15 | 89.58 | 83.97 | 63.75 | 77.76 | 61.55 | 71.93 | 60.76 | 58.92 | 51.95 | 54.65 | 62.43 |
| ✓ | ✓ | 96.54 | 89.07 | 95.69 | 92.15 | 84.61 | 69.46 | 79.95 | 65.21 | 74.23 | 66.05 | 66.33 | 58.81 | 79.27 | 75.85 |

## 5.3 Further Analysis

**More angle visualizations.** In Figure 5, we show the effectiveness of local adaptation in FedGELA and verify the convergence of fixed classifier as ETF and local adaptation compared with FedAvg. Together with Figure 3, it can be seen that, compared with FedAvg, both FedGE and FedGELA converge faster to a larger angle between all class means in global. In the meanwhile, the angle between existing classes of FedGELA in the local is much larger, which proves FedGELA converges better than FedAvg and the adaptation brings little limits to convergence but many benefits to local performance under different levels of PCDD.

**Hyper-parameter.** FedGELA introduces constrain $E_H$ on the features and the length $E_W$ of classifier vectors. We perform $L_2$ norm on all features in FedGELA, which means $E_H = 1$. For the length of the classifier, we tune it as hyper-parameter. As shown in Figure 4, from a large range from 10e3 to 10e6 of $E_W$, our method achieves bilateral improvement compared to FedAvg on all datasets.

**Ablation studies.** Since our method includes two parts: global ETF and local adaptation, we illustrate the average accuracy of FedGELA on all splits of SVHN, CIFAR10/100, and Fed-ISIC2019 without the global ETF or the local adaptation or both. As shown in Table 4, only adjusting the local classifier does not gain much in personal or global tasks, and compared with FedGE, FedGELA achieves similar generic performance on the four datasets but much better performance on the personal tasks.

Table 5: Performance of FedGELA compared with FedAvg and the best baseline under pure PCDD settings on CIFAR10 and SVHN datasets. $P\varrho C\varsigma$ means that the dataset is divided into $\varrho$ clients and each client has $\varsigma$ classes. We show the improvement in red on each baseline compared to FedGELA.

| Dataset (split) | Metric | FedAvg | Best Baseline | FedGELA |
|---|---|---|---|---|
| CIFAR10(P10C2) | PA | 92.08+3.76 | 94.07+1.77 | 95.84 |
| | GA | 47.26+12.34 | 52.02+7.58 | 59.60 |
| CIFAR10(P50C2) | PA | 91.74+3.68 | 93.22+2.20 | 95.42 |
| | GA | 36.22+18.56 | 44.74+10.04 | 54.78 |
| SVHN(P10C2) | PA | 95.64+3.11 | 97.02+1.73 | 98.75 |
| | GA | 69.34+14.22 | 76.06+7.50 | 83.56 |
| SVHN(P50C2) | PA | 94.87+3.50 | 96.88+1.49 | 98.37 |
| | GA | 66.94+10.24 | 72.97+4.21 | 77.18 |

Table 6: Performance of choosing different $\Phi$. Assuming the row vector of distribution matrix$(\Phi_k)_c^T$ is related to class distribution $\frac{n_{k,c}}{n_k}$ and the relationship as $Q_k(\frac{n_{k,c}}{n_k})$. Except for $Q_k(x) = x$, we have also considered employing alternative methods like employing an exponential $Q_k(x) = e^x$ or power function $Q_k(x) = x^{\frac{1}{2}}$ of the number of samples.

| Dataset (split) | Metric | $Q_k(x) = e^x$ | $Q_k(x) = x^{\frac{1}{2}}$ | $Q_k(x) = x$(ours) |
|---|---|---|---|---|
| SVHN(IID) | PA | 95.12 | 95.43 | 94.84 |
| | GA | 94.32 | 93.99 | 94.66 |
| SVHN($\beta = 0.5$) | PA | 96.18 | 95.56 | 96.27 |
| | GA | 93.28 | 93.22 | 93.66 |
| SVHN($\beta = 0.1$) | PA | 98.33 | 98.21 | 98.52 |
| | GA | 78.95 | 77.18 | 78.88 |

**Performance under pure PCDD setting.** To verify our method under pure PCDD, we decouple the PCDD setting and the ordinary heterogeneity (Non-PCDD). In Table 5, we use PxCy to denote the dataset is divided in to x clients with y classes, and in each round, 10 clients are selected into federated training. The training round is 100. According to the results, FedGELA achieves significant improvement especially $18.56\%$ to FedAvg and $10.04\%$ to the best baseline on CIFAR10 (P50C2).

**Other types of $\Phi$.** Considering the aggregation of local classifiers should align with the global classifier, which ensures the validity of theoretical analyses from both global and local perspectives, $\sum_{k=1}^{N} p_k \Phi_k$ should be $\mathbf{1}$ ($\mathbf{1}$ is all-one matrix). Assuming the row vector of distribution matrix$(\Phi_k)_c^T$ is related to class distribution $\frac{n_{k,c}}{n_k}$ and the relationship as $Q_k(\frac{n_{k,c}}{n_k})$. The equation can be rewrite as: $\gamma \sum_{k=1}^{N} p_k Q_k(\frac{n_{k,c}}{n_k}) = \mathbf{1}$, where $\gamma$ is the scaling constant. In our FedGELA, to avoid sharing statistics for privacy, we only find one potential way that $Q_k(\frac{n_{k,c}}{n_k}) = \frac{n_{k,c}}{n_k}$ and $\gamma = \frac{1}{C}$. In this part, we have also considered employing alternative methods like employing an exponential or power function of the number of samples. As shown in the Table 6, other methods need to share $Q_k(\frac{n_{k,c}}{n_k})$ but achieve the similar performance compared to FedGELA, which exhibits the merit of our choice.

In Appendix D, we provide more experiments from other perspectives like communication efficiency and the local burden of storing and computation, to show promise of FedGELA.

## 6 Conclusion

In this work, we study the problem of *partially class-disjoint data* (PCDD) in federated learning on both personalized federated learning (P-FL) and generic federated learning (G-FL), which is practical and challenging due to the angle collapse of classifier vectors for the global task and the waste of space for the personal task. We propose a novel method, FedGELA, to address the dilemma via a bilateral curation. Theoretically, we show the local and global convergence guarantee of FedGELA and verify the justification on the angle of global classifier vectors and on the angle between locally existing classes. Empirically, extensive experiments show that FedGELA achieves promising improvements on FedAvg under PCDD and outperforms state-of-the-art methods in both P-FL and G-FL.

## Acknowledgement

The work is supported by the National Key R&D Program of China (No. 2022ZD0160702), STCSM (No. 22511106101, No. 22511105700, No. 21DZ1100100), 111 plan (No. BP0719010) and National Natural Science Foundation of China (No. 62306178). Ziqing Fan and Ruipeng Zhang were partially supported by Wu Wen Jun Honorary Doctoral Scholarship, AI Institute, Shanghai Jiao Tong University. Bo Han was supported by the NSFC Young Scientists Fund No. 62006202, NSFC General Program No. 62376235, Guangdong Basic and Applied Basic Research Foundation No. 2022A1515011652, and CCF-Baidu Open Fund.

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
