## A    Neural Collapse and simplex ETF

Neural collapse [26] is an intuitive observation that happens at the terminal phase of a well-trained model on a balanced dataset that last-layer features converge to within-class mean, and all within-class means and their corresponding classifier vectors converge to ETF as shown in Figure 6. The main results can be concluded as follows:

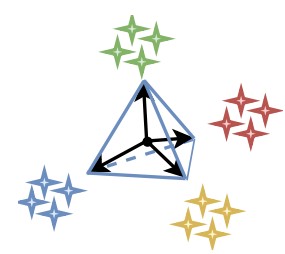

- (NC1) Variability of the last-layer features $\Sigma := \mathrm{Avg}_{i,c}\{(h_c^i - h_c)(h_c^i - h_c)^T\}$ collapse within-class: $\Sigma \to \mathbf{0}$, where $h_c^i$ is the last-layer feature of the $i$-th sample in the $c$-th class, and $h_c$ is the within-class mean of c-th class's features.

- (NC2) Convergence to a simplex ETF. Last-layer features converge to within-class mean, and all within-class means and their corresponding classifier vectors converge to a simplex ETF.

- (NC3) Self duality: $\tilde{h}_c = \mathbf{W}_c/\|\mathbf{W}_c\|$, where $\tilde{h}_c = (h_c - \overline{h})/\|h_c - \overline{h}\|$ and $\mathbf{W}_c$ is the classifier vector of the $c$-th class.

- (NC4) Simplification to the nearest class center prediction: $\mathrm{argmax}_c \langle h, \mathbf{W}_c \rangle = \mathrm{argmin}_c \|h - h_c\|$, where $h$ is the last-layer feature.

Figure 6: ETF structure. Stars with different colors denote features of different classes and black arrows denote the classifier vector for each class.

**Lemma 1** (ETF). *When solving objective defined in Eq* (6) *in balanced C-class classification tasks with LPM and CE loss, neural collapse merges, which means $\forall 1 \leqslant i \leqslant n_c, 1 \leqslant c \leqslant C$, last layer features $H_i^*$ and corresponding classifier $\mathbf{W}_j^*$ converge as:*

$$\frac{h_c^{i,*}}{\sqrt{E_H}} = \frac{\mathbf{W}_c^*}{\sqrt{E_W}} = m_c^*,$$

*where $m_c^*$ forms a simplex equiangular tight frame (ETF) defined as:*

$$\mathbf{M} = \sqrt{\frac{C}{C-1}} U \left( \mathbf{I}_C - \frac{1}{C} \mathbf{1}_C \mathbf{1}_C^T \right),$$

*where $\mathbf{M} = [m_1^*, \cdots, m_C^*] \in \mathbb{R}^{d \times C}, U \in \mathbb{R}^{d \times C}$ allows a rotation and satisfies $U^T U = \mathbf{I}_C$ and $\mathbf{1}_C$ is an all-ones vector.*

To analyze this phenomenon, some studies simplify deep neural networks as last-layer features and classifier (layer-peeled model)[9, 12, 40, 53] with proper constraints or regularizations. In the view of layer-peeled model (LPM), training $\mathbf{W}$ with constraints on the weights can be seen as training the C-class classification head $\mathbf{W}^L = \{\mathbf{W}_1, ..., \mathbf{W}_C\}$ and features $H = \{h^1, ..., h^N\}$ of all $n$ samples output by last layer of backbone with constraints $E_W$ and $E_H$ respectively. Therefore, $\forall 1 \leqslant c \leqslant C, 1 \leqslant i \leqslant N$, the training objective with commonly used cross-entropy loss can be described as:

$$\min_{H, \mathbf{W}^L} \frac{1}{n} \sum_{i=1}^{N} \mathcal{L}_{CE}\left(h^i, \mathbf{W}\right),$$

$$\text{s.t. } \left\|\mathbf{W}_c^L\right\|^2 \leqslant E_W, \ \left\|h^i\right\|^2 \leqslant E_H. \tag{6}$$

In the balanced dataset, as described in Lemma 1, any solutions to this model merge neural collapse and form a simplex equiangular tight frame (ETF), which means ETF is optimal classifier in the balanced case of LPM.

**Lemma 2** (Fixing classifier as ETF). *No matter dataset is balanced or imbalanced, fixing the classification head as ETF with scaling length of $\sqrt{E_W}$ in the layer-peeled model and optimizing the following objective:*

$$\min_{H} \frac{1}{n} \sum_{i=1}^{n} \mathcal{L}_{CE}\left(h^i, \sqrt{E_W}\mathbf{M}\right),$$

$$\text{s.t. } \left\|h^i\right\|^2 \leqslant E_H, \forall 1 \leqslant i \leqslant n.$$

*Then the same solution in the Lemma 1 is obtained.*

Table 7: Notations and their corresponding meanings.

| Notation | Meaning |
|---|---|
| $F$ | global loss function |
| $F_k$ | local loss function of client k |
| $\tau$ | local iteration in a curtain round |
| $b_k^\tau$ | mini-bath of a certain iteration |
| $H$ | last layer features |
| $H^i$ | last layer feature of i-th sample |
| $H_k$ | last layer features of k-th client |
| $H_{k,c}^i$ | last layer feature of i-th sample of c-th class in k-th client |
| $\mathbf{M}$ | ETF matrix |
| $m_j$ | j-th classifier vector in ETF |
| $\phi$ | set of adjusted matrices |
| $\phi_k$ | adjusted matrix of client k |
| $t$ | number of communication rounds |
| $\phi_{k,c}$ | adjusted weight of c-th class in client k |
| $p_k$ | sample fraction of client k |
| $N$ | number of clients |
| $\mathbf{W}$ | total model |
| $\mathbf{W}_k$ | total model of client k |
| $\mathbf{W}_k^{tE}$ | model of client k after E-1 aggregation with $\tau$ additional iteration |
| $\mathbf{W}_k^{tE+\frac{1}{2}}$ | model after aggregation of client k in round t |
| $\mathbf{W}_k^{\tau E+\frac{1}{2}\phi_k}$ | model after aggregation and adjusted of client k in round t |
| $\mathbf{W}_g$ | global model |
| $\mathbf{W}^L$ | classifier |
| $\mathbf{W}^{-L}$ | backbone |
| $\mathbf{W}_k^L$ | classifier of k-th client |
| $\mathbf{W}_k^{-L}$ | backbone of k-th client |
| $\mathbf{W}_{k,c}^L$ | classifier of c-th class in k-th client |
| $\mathbf{W}_{k,c}^{-L}$ | backbone of c-th class in k-th client |
| $g$ | gradient |
| $n$ | number of samples |
| $n_k$ | sample number of client k |
| $n_{k,c}$ | sample number of c-th class in client k |

As shown in Lemma 2, recent studies prove that no matter dataset is balanced or not, by fixing the classifier as a randomly ETF with scaling $\sqrt{E_W}$ and constraining on last layer features, LPM can reach the optimal structure as described in Lemma 1. We also prove in Theorem 1 that by fixing the classifier of all clients as ETF, in the strongly convex case, the global model can also reach the condition as Lemma 1 which meets the requirement of G-FL.

# B Implementation of Theoretical Analysis

## B.1 Notations and Assumptions

Before starting our proof, we pre-define some notations and assumptions used in the following lemmas and theorems. First, we make the assumptions on loss functions $F_1, F_2, \cdots, F_N$ of all clients and their gradient functions $\nabla F_1, \nabla F_2, \cdots, \nabla F_N$. We use $tE + \tau$ to denote $\tau$-th local iteration in round $t$, $tE$ to denote the state that just finishing local training, $tE + \frac{1}{2}$ to denote the stage after aggregation and $tE + \frac{1}{2}\mathbf{\Phi}$ to denote the stage after local adaptation. In Assumption 1 and Assumption 2, we characterize the smoothness, bound on the variance of stochastic gradients and convexity of each $F_N$. In Assumption 3, the norm of stochastic gradients is bounded, which is commonly used in many FL algorithms together with Assumption 2 to prove the global convergence [8, 18]. An existing study points out that there is a contradiction between them [24, 32]. Therefore, we show the concrete assumption description in Assumption 5 and convergence guarantee without bounded norm of stochastic gradients in Theorem 3 and Theorem 4. In Assumption 4, the heterogeneity is reflected in the distance between local optimum $W_k^*$ and global optimum $\mathbf{W}^*$ and the loss deviation before and after aggregation, which is bounded by $\Gamma_1$ and $\Gamma_2$ respectively.

**Assumption 1** (L-smooth and bounded variance of stochastic gradients). $F_1, \cdots, F_N$ are L-smooth:

$$\forall u, \forall v, 1 \leq k \leq N, F_k(u) \leq F_k(v) + (u-v)^T \nabla F_k(v) + \frac{L}{2}\|u-v\|_2^2,$$

and their variance of stochastic gradients is bounded:

$$\forall t \geq 0, 1 \leq k \leq N, \frac{1}{2} \leq \tau \leq E, \mathbb{E}\|\nabla F_k(W_k^{tE+\tau}, \xi_k^{tE+\tau}) - \nabla F_k(W_k^{tE+\tau})\|^2 \leq \sigma_k^2.\xi = \{\mathbf{x}, y\} \tag{7}$$

**Assumption 2** ($\mu$-strongly convex). $F_1, \cdots, F_N$ are $\mu$-strongly convex:

$$\forall u, \forall v, 1 \leq k \leq N, F_k(u) \geq F_k(v) + (u-v)^T \nabla F_k(v) + \frac{\mu}{2}\|u-v\|_2^2 \tag{8}$$

**Assumption 3** (Bounded norm of stochastic gradients). *The expected squared norm of stochastic gradients is bounded:*

$$\forall t \geq 0, 1 \leq k \leq N, \frac{1}{2} \leq \tau \leq E, \mathbb{E}\left\|\nabla F_k(\mathbf{W}_k^{tE+\tau}, b_k^{tE+\tau})\right\|^2 \leq G^2.$$

**Assumption 4** (Bounded heterogeneity). *The deviation between local and global optimum and the deviation between local and global loss before and after aggregation are both bounded:*

$$\forall t \geq 0, 1 \leq k \leq N, \|\mathbf{W}_k^* - \mathbf{W}^*\| \leq \Gamma_1 \;\;\&\;\; \|\nabla F_k^{tE} - \nabla F_k^{tE+\frac{1}{2}}\|_2 \leq \Gamma_2.$$

**Assumption 5** (Correct bounded norm of stochastic gradients [24, 32]). *Let Assumptions 1 and 2 hold. Then the expected squared norm of the stochastic gradient is bounded by:*

$$\mathbb{E}\|\nabla F_k(\mathbf{W}_k^{tE+\tau}, \xi_k^{tE+\tau})\|^2 \leq 4L\kappa \left[F_k(\mathbf{W}_k^{tE+\tau}) - F_k(\mathbf{W}_k^*)\right] + G_k,$$

$$\text{where } \kappa = \frac{L}{\mu} \text{ and } G_k = 2\mathbb{E}\|\nabla F_k(\mathbf{W}_k^*, \xi_k^{tE+\tau})\|^2$$

**Lemma 3** (Results of one step SGD [33]). *Let Assumption 1 hold. From the beginning of communication round $t+1$ to the last local update step, the loss function of an arbitrary client can be bounded as:*

$$\mathbb{E}[\mathcal{F}_k^{(t+1)E}] \leq \mathcal{F}_k^{tE+\frac{1}{2}\phi_k} - (\eta - \frac{L\eta^2}{2}) \sum_{e=\frac{1}{2}\phi_k}^{E-1} \|\nabla \mathcal{F}_k^{tE+e}\|_2^2 + \frac{LE\eta^2}{2}\sigma^2.$$

**Lemma 4** (Results of one step SGD [8, 20]). *Assume Assumption 1 holds. If $\eta_t \leq \frac{1}{4L}$, we have*

$$\mathbb{E}\|W^{t+1} - W^\star\|^2 \leq (1 - \eta_t\mu)\mathbb{E}\|W^t - W^\star\|^2 + 6L\eta_t^2\Gamma$$

$$+ \eta_\tau^2\mathbb{E}\|\mathbf{g}_\tau - \overline{\mathbf{g}}_\tau\|^2 + 2\mathbb{E}\sum_{k=1}^{N} p_k\|W^t - W_k^{tE}\|^2,$$

*where $\Gamma = F^* - \sum_{k=1}^{N} p_k F_k^\star \geq 0$*

**Lemma 5** (Math tool from Stich [30]). *Assume there are two non-negative sequences $\{r_\tau\}, \{s_\tau\}$ that satisfy the relation*

$$r_{\tau+1} \leq (1 - \alpha\gamma_\tau) r_\tau - b\gamma_\tau s_\tau + c\gamma_\tau^2$$

*for all $\tau \geq 0$ and for parameters $b > 0, a > 0, c > 0$ and non-negative step sizes $\{\gamma_\tau\}$ with $\gamma_\tau \leq \frac{1}{d}$ for a parameter $d \geq a, d > 0$. Then, there exists weights $\omega_\tau \geq 0, W_T := \sum_{\tau=0}^{T} \omega_\tau$, such that:*

$$\frac{b}{W_T}\sum_{\tau=0}^{T} s_\tau\omega_\tau + ar_{T+1} \leq 32dr_0\exp\left[-\frac{aT}{2d}\right] + \frac{36c}{aT}$$

**Lemma 6** (Bounding the variance [8, 20]). *Assume Assumption 1 holds. It follows that*

$$\mathbb{E}\left[\|\mathbf{g}_\tau - \overline{\mathbf{g}}_\tau\|^2\right] \leq \sum_{k=1}^{N} p_k^2\sigma_k^2.$$

**Lemma 7.** *(Bounding the divergence of $\{W_k^{tE}\}$ [20].). Assume Assumption 3, that $\eta_t$ is non-increasing and $\eta_t \leq 2\eta_{t+E}$ for all $t \geq 0$. It follows that:*

$$\mathbb{E}\left[\sum_{k=1}^{N} p_k\|\mathbf{W}^t - \mathbf{W}_k^{tE}\|^2\right] \leq 4\eta_t^2(E-1)^2G^2 + \sum_{k=1}^{N} p_k\|\mathbf{\Phi}_k\mathbf{W}^L - \mathbf{W}^L\|$$

*Proof.* Different from the Lemma in [31], we consider the ETF structure in $W$. Therefore, for any $t > 0$ and $k = 1, 2, \cdots, N$, we use the fact that $\eta_t$ is non-increasing and $\eta_{tE} \leq 2\eta_t$, then

$$\mathbb{E}\sum_{k=1}^{n} p_k\|\mathbf{W}^t - \mathbf{W}_k^{tE}\|^2 = \mathbb{E}\sum_{k=1}^{N} p_k(\|\mathbf{W}^t - \mathbf{W}_k^{tE}\|^2 + \|\mathbf{\Phi}_k\mathbf{W}^L - \mathbf{W}^L\|^2) \tag{9}$$

$$\underset{\text{SGD}}{\leq} \sum_{k=1}^{N} p_k\left(\mathbb{E}\sum_{\tau=2}^{E}(E-1)\|\eta_\tau\nabla F_k(\mathbf{W}_k^{t\tau}, \xi_k^{t\tau})\|^2 + \|\mathbf{\Phi}_k\mathbf{W}^L - \mathbf{W}^L\|^2\right) \tag{10}$$

$$\underset{\text{Assumption 3}}{\leq} \sum_{k=1}^{N} p_k\left(\eta_{tE}^2(E-1)^2G^2 + \|\mathbf{\Phi}_k\mathbf{W}^L - \mathbf{W}^L\|^2\right) \tag{11}$$

$$\underset{\eta_{tE} \leq 2\eta_t}{\leq} 4\eta_t^2(E-1)^2G^2 + \sum_{k=1}^{N} p_k\|\mathbf{\Phi}_k\mathbf{W}^L - \mathbf{W}^L\|. \tag{12}$$

$\square$

## B.2 Proof of Theorem 1

*Proof.* Similar to [20], from Lemma 4, Lemma 7 and Lemma 6, let $\gamma = \max\{8\kappa, E\}$ and $\eta_{tE} \leq 2\eta_t$, it follows that

$$\mathbb{E}[F(\mathbf{W}_g)] - F(\mathbf{W}^*) \leq \frac{\kappa}{\gamma + T - 1}\left(\frac{2B}{\mu} + \frac{\mu\gamma}{2}\mathbb{E}\|\mathbf{W}^1 - \mathbf{W}^*\|^2\right),$$

which uses the same proof technique in [20]. And we apply the cross-entropy loss for $F$, then $F(\mathbf{W}) = -\log[(\mathbf{W}^L)^T h_c^i]$ for class $c$ on sample $i$. Then we have

$$\mathbb{E}[F(\mathbf{W}_g)] - F(\mathbf{W}^*) = \mathbb{E}\left[\log\frac{(\mathbf{W}^{L,*})^T h_c^{i,*}}{(\mathbf{W}_g^L)^T h_c^i}\right].$$

So Theorem 1 is proved.

$\square$

## B.3 Proof of Theorem 2

*Proof.* We start our proof from one step of SGD defined in Lemma 3:

$$\mathbb{E}[F_k^{(t+1)E}] \leq F_k^{tE+\frac{1}{2}\phi_k} - (\eta - \frac{L\eta^2}{2}) \sum_{e=\frac{1}{2}\phi_k}^{E-1} \|\nabla F_k^{tE+e}\|_2^2 + \frac{LE\eta^2}{2}\sigma^2.$$

We can take apart the $F_k^{tE+\frac{1}{2}\Phi_k}$ and have the fact that:

$$\begin{aligned}
\|F_k^{tE+\frac{1}{2}\Phi_k}\| &= \left\|F_k^{tE} + F_k^{tE+\frac{1}{2}\Phi_k} - F_k^{tE}\right\| \\
&\leqslant \left\|F_k^{tE}\right\| + \left\|F_k^{tE+\frac{1}{2}\Phi_k} - F_k^{tE}\right\| \\
&\leqslant \|F_k^{tE}\| + \|F_k^{tE+\frac{1}{2}\Phi_k} - F_k^{tE+\frac{1}{2}} + F_k^{tE+\frac{1}{2}} - F_k^{tE}\| \\
&\leqslant \|F_k^{tE}\| + \left\|F_k^{tE+\frac{1}{2}\Phi_k} - F_k^{tE+\frac{1}{2}}\right\| + \|F_k^{tE+\frac{1}{2}} - F_k^{tE}\| \\
&\leqslant \|F_k^{tE}\| + L\|\mathbf{W}_k^{tE+\frac{1}{2}\Phi_k} - \mathbf{W}_k^{tE+\frac{1}{2}}\| + \Gamma_1 \\
&= \|F_k^{tE}\| + L\|\Phi_k\mathbf{W}^L - \mathbf{W}^L\| + \Gamma_1
\end{aligned}$$

Take it back to the original equation, therefore we have the:

$$\mathbb{E}[F_k^{(t+1)E}] \leq \|F_k^{tE}\| - (\eta - \frac{L\eta^2}{2}) \sum_{e=\frac{1}{2}\phi_k}^{E-1} \|\nabla F_k^{tE+e}\|_2^2 + \frac{LE\eta^2}{2}\sigma^2 + L\|\Phi_k\mathbf{W}^L - \mathbf{W}^L\| + \Gamma,$$

By applying Assumption 3 that:$\mathbb{E}\|\nabla F_k(u, b_k^\tau)\|^2 \leq G^2$, results will be:

$$\mathbb{E}[F_k^{(t+1)E}] \leqslant F_k^{tE} - (\eta - \frac{L}{2}\eta^2)EG^2 + \frac{LE\eta^2}{2}\sigma^2 + L\|\Phi_k\mathbf{W}^L - \mathbf{W}^L\| + \Gamma.$$

Here we complete our proof. $\qquad\square$

## B.4 Contradictory of the Assumptions and Correction

### B.4.1 Contradictory on Assumption 3.

We will prove that if Assumptions 1 and 2 hold, the stochastic gradients cannot be uniformly bounded.

*Proof.* If Assumptions 1 and 2 hold, which means $F_k$ is both *L-smooth* and *$\mu$-strong convex*, we have:

$$2\mu[F_k(\mathbf{W}) - F_k(\mathbf{W}^*)] \leq ||\nabla F_k(\mathbf{W})||^2 \tag{13}$$

The proof of (13) will be given below. And under the false stochastic gradients uniformly bounded assumption 3, we have $\mathbb{E}[||\nabla F_k(\mathbf{W}_k^{tE+\tau}, \xi_k^{tE+\tau})||^2] \leq G^2$. So we get

$$\begin{aligned}
2\mu[F_k(\mathbf{W}) - F_k(\mathbf{W}^*)] &\leq ||\nabla F_k(\mathbf{W})||^2 \\
&\leq ||\mathbb{E}[\nabla F_k(\mathbf{W}_k^{tE+\tau}, \xi_k^{tE+\tau})]||^2 \\
&\leq \mathbb{E}[||\nabla F_k(\mathbf{W}_k^{tE+\tau}, \xi_k^{tE+\tau})||^2] \\
&\leq G^2
\end{aligned} \tag{14}$$

Therefore, we have the result that $F_k(\mathbf{W}) - F_k(\mathbf{W}^*) \leq \frac{G^2}{2\mu}$. Using the strong convex in (8) with $\mathbf{W} = \mathbf{W}^*$ that $\nabla F_k(\mathbf{W}^*) = 0$, we will have:

$$F_k(\mathbf{v}) - F_k(\mathbf{W}^*) \geq (\mathbf{v} - \mathbf{W}^*)^T \nabla F_k(\mathbf{W}^*) + \frac{\mu}{2}||\mathbf{v} - \mathbf{W}^*||^2 = \frac{\mu}{2}||\mathbf{v} - \mathbf{W}^*||^2 \tag{15}$$

By combining (15) and (14), it follows that

$$\frac{G^2}{2\mu} \geq F_k(\mathbf{W}) - F_k(\mathbf{W}^*) \geq \frac{\mu}{2}||\mathbf{W} - \mathbf{W}^*||^2,$$

$$||\mathbf{W} - \mathbf{W}^*||^2 \leq \frac{G^2}{\mu^2}. \tag{16}$$

where (16) is clearly wrong for sufficiently large $||\mathbf{W} - \mathbf{W}^*||^2$.

$\qquad\square$

### B.4.2 Proof of corrected Assumption 5.

*Proof.* Note that:

$$||a||^2 = ||a - b + b||^2 \leq 2||a - b||^2 + 2||b||^2 \tag{17}$$

$$\implies \frac{1}{2}||a||^2 - ||b||^2 \leq ||a - b||^2 \tag{18}$$

If Assumptions 1 and 2 hold, combined with (18) we have:

$$\frac{1}{2}\mathbb{E}[||\nabla F_k(\mathbf{W}_k^{tE+\tau}, \xi_k^{tE+\tau})||^2] - \mathbb{E}[||\nabla F_k(\mathbf{W}_k^*, \xi_k^{tE+\tau})||^2]$$

$$= \mathbb{E}\left[\frac{1}{2}||\nabla F_k(\mathbf{W}_k^{tE+\tau}, \xi_k^{tE+\tau})||^2 - ||\nabla F_k(\mathbf{W}_k^*, \xi_k^{tE+\tau})||^2\right]$$

$$\leq \mathbb{E}\left[||\nabla F_k(\mathbf{W}_k^{tE+\tau}, \xi_k^{tE+\tau}) - \nabla F_k(\mathbf{W}_k^*, \xi_k^{tE+\tau})||^2\right]$$

$$\underset{\text{Eq.(7)}}{\leq} L^2||\mathbf{W}_k^{tE+\tau} - \mathbf{W}_k^*||^2$$

$$\underset{\text{Eq.(8)}}{\leq} \frac{2L^2}{\mu}[F_k(\mathbf{W}_k^{tE+\tau}, \xi_k^{tE+\tau}) - F_k(\mathbf{W}_k^*, \xi_k^{tE+\tau})]$$

$$= 2L\kappa[F_k(\mathbf{W}_k^{tE+\tau}, \xi_k^{tE+\tau}) - F_k(\mathbf{W}_k^*, \xi_k^{tE+\tau})]$$

So we get: $\mathbb{E}[||\nabla F_k(\mathbf{W}_k^{tE+\tau}, \xi_k^{tE+\tau})||^2] \leq 4L\kappa[F_k(\mathbf{W}_k^{tE+\tau}) - F_k(\mathbf{W}_k^*)] + G_k.$ □

### B.4.3 Correction.

In this part, we provide convergence results without the bounded norm of stochastic gradient defined in Assumption 3. In Theorem 3 and Theorem 4, we show the corrected results of global and local convergence, respectively.

**Theorem 3** (Global Convergence). *If $F_1, ..., F_N$ are all $L$-smooth, $\mu$-strongly convex, and the variance and norm of $\nabla F_1, ..., \nabla F_N$ are bounded by $\sigma$ and $G$. Choose $\kappa = L/\mu$ and $\gamma = \frac{32}{k(\mu-k)}L^2\kappa(E-1)^2 - 1$, for all classes $c$ and sample $i$, expected global representation by cross-entropy loss will converge to:*

$$\mathbb{E}\left[\log \frac{(\mathbf{W}^{L,*})^T h_c^{i,*}}{(\mathbf{W}_g^L)^T h_c^i}\right] \leq \frac{\kappa}{\gamma + T - 1}\left(\frac{2B}{\mu} + \frac{\mu\gamma}{2}\mathbb{E}||\mathbf{W}^1 - \mathbf{W}^*||^2\right),$$

*where in FedGELA, $B = \sum_{k=1}^N(p_k^2\sigma^2 + p_k||\mathbf{\Phi}_k\mathbf{W}^L - \mathbf{W}^L||) + 6L\Gamma_1 + 8(E-1)^2G^2$ and $G = \sum_{k=1}^K p_k G_k = 2\sum_{k=1}^K p_k\mathbb{E}[||\nabla F_k(\mathbf{W}_k^*, \xi_k^{L+\tau})||^2]$. Since $\mathbf{W}^L = \mathbf{W}^{L,*}$ and $(\mathbf{W}^{L,*})^T h_{c_i}^{i,*} \geq \mathbb{E}[(\mathbf{W}^L)^T h_{c_i}^i]$, $h_{c_i}^i$ will converge to $h_{c_i}^{i,*}$.*

Similar to Theorem 1, the variable $B$ in Theorem 3 represents the impact of algorithmic convergence $(p_k^2\sigma^2)$, non-iid data distribution $(6L\Gamma_1)$, and stochastic optimization $(8(E-1)^2G^2)$. The only difference between FedAvg, FedGE, and our FedGELA lies in the value of $B$ while others are kept the same. FedGE and FedGELA have a smaller $G$ compared to FedAvg because they employ a fixed ETF classifier that is predefined as optimal. FedGELA introduces a minor additional overhead $(p_k||\mathbf{\Phi}_k\mathbf{W}^L - \mathbf{W}^L||)$ on the global convergence of FedGE due to the incorporation of local adaptation to ETFs. The cost might be negligible, as $\sigma$, $G$, and $\Gamma_1$ are defined on the whole model weights while $p_k||\mathbf{\Phi}_k\mathbf{W}^L - \mathbf{W}^L||$ is defined on the classifier. To verify this, we conduct experiments in Figure 3(a), and as can be seen, FedGE and FedGELA have similar quicker speeds and larger classification angles than FedAvg.

**Theorem 4** (Local Convergence). *If $F_1, ..., F_N$ are $L$-smooth and the heterogeneity is bounded by $\Gamma_2$, clients' expected local loss satisfies:*

$$F_K(\mathbf{W}_k^{tE+\frac{1}{2}\mathbf{\Phi}}) - F_k^*(\mathbf{W}_k^*) \leq L||\mathbf{W}_k^{tE+\frac{1}{2}} - \mathbf{W}^*|| + D,$$

*where in FedGELA, $D = \Gamma_2 + ||\mathbf{\Phi}_k\mathbf{W}^L - \mathbf{W}^L||E_w$, which means the local convergence is highly related to global convergence and bounded by D.*

In Theorem 4, only "D" is different on the convergence among FedAvg, FedGE, and FedGELA. The local convergence is highly related to global convergence and bounded by D. Adapting the local classifier will introduce additional cost of $L \left\| \mathbf{\Phi}_k \mathbf{W}^L - \mathbf{W}^L \right\|$, which might limit the speed of local convergence. However, FedGELA might reach better local optimal by adapting the feature structure. As introduced and verified in Figure 3 (c) in the main pape, the adapted structure expands the decision boundaries of existing major classes and better utilizes the feature space wasted by missing classes.

*Proof.* We can prove the theorem by inserting $\mathbf{W}_k^*$ and taking apart the local loss function:

$$
\begin{aligned}
& F_k(\mathbf{W}_k^{tE+\frac{1}{2}\phi}) - F_k^*(\mathbf{W}_k^*) \\
& \leqslant \|F_k(\mathbf{W}_k^{tE+\frac{1}{2}\mathbf{\Phi}}) - F_k(\mathbf{W}^*)\| + \|F_k(\mathbf{W}^*) - F_k^*(\mathbf{W}_k^*)\| \\
& \leq L\|\mathbf{W}_k^{tE+\frac{1}{2}\mathbf{\Phi}} - \mathbf{W}^*\| + \Gamma_2 \\
& = L\|\mathbf{W}_k^{tE+\frac{1}{2}} - \mathbf{W}^*\| + \Gamma_2 + \|\mathbf{\Phi}_k\mathbf{W}^L - \mathbf{W}^L\|
\end{aligned}
$$

Here we complete the proof. The last and the second last inequalities are derived from L-smooth and bounded heterogeneity defined in Assumption 1 and Assumption 4 respectively. □

### B.5 Implementation of the Justification Experiments.

To verify the contradiction of the local objective and global objective, we track the angle of classifier vectors between locally existing classes and locally missing classes in an individual client. We denote "existing angle" as the angle of classifier vectors belonging to classes that exist in a local client while "missing angle" is the angle of classifier vectors belonging to non-existing classes. In Sec. 3.2 and shown in Figure 2, the tracking experiment is conducted on CIFAR10 with 10 clients under Dir ($\beta = 0.1$). To verify the effectiveness and convergence of FedGELA, we track the angle between class means of locally existing classes and all classes in local and global, respectively. In Sec. 4 and illustrated in Figure 3, the tracking experiment is conducted on CIFAR10 with 50 clients under Dir ($\beta = 0.2$). In the experiment, we also provide more results under different situations illustrated in Figure 5.

## C  Implementation of the Experiment

### C.1  Model

Resnet18 backbone [8, 17, 25, 33, 42, 47, 49] is commonly used in many federated experiments on CIFAR10 and CIFAR100 datasets, here we also use it as the backbone for SVHN, CIFAR10 and CIFAR100. Since there are many algorithms that are feature-based like MOON and FedProto, therefore we use one layer of FC as the projection layer (the hidden size is 84 for SVHN and CIFAR10 and 512 for CIFAR100) followed by classification head. For FedGELA and FedGE, the model is a backbone, projection layer with a simplex ETF or adapted ETF. For Fed-ISIC2019, we follow the setting of Flambly and use pre-trained Efficientnet b0 with the same projection layer (the hidden size is 84) as the model.

### C.2  Partition Strategy

Dirichlet distribution (Dir ($\beta$)) is practical and commonly used in FL settings [3, 8, 17, 21, 25, 33, 47]. As in many recent works, we deploy $Dir$ ($\beta = 10000$) to simulate the almost IID situations and $Dir$ ($\beta = 0.5, 0.2, 0.1$) to simulate the different levels of Non-IID situations. As shown in Figure 7, we provide the data distribution heatmap among clients of SVHN, CIFAR10, and CIFAR100 under Dirichlet distribution with different $\beta$. It can be seen that Dirichlet distribution can also generate practical PCDD data distribution. We also provide the data distribution heatmap of Fed-ISIC2019 shown in Figure C.2. In Fed-ISIC2019, there exists a true PCDD situation that needs to be solved. To verify full participating (10,10) and straggler situations when client numbers are increasing, we split SVHN, CIFAR10, and CIFAR100 into 10 and 50 clients, and in each round, 10 clients are randomly selected into federated training. In FedISIC2019, there are 6 clients with 8 classes of samples and we

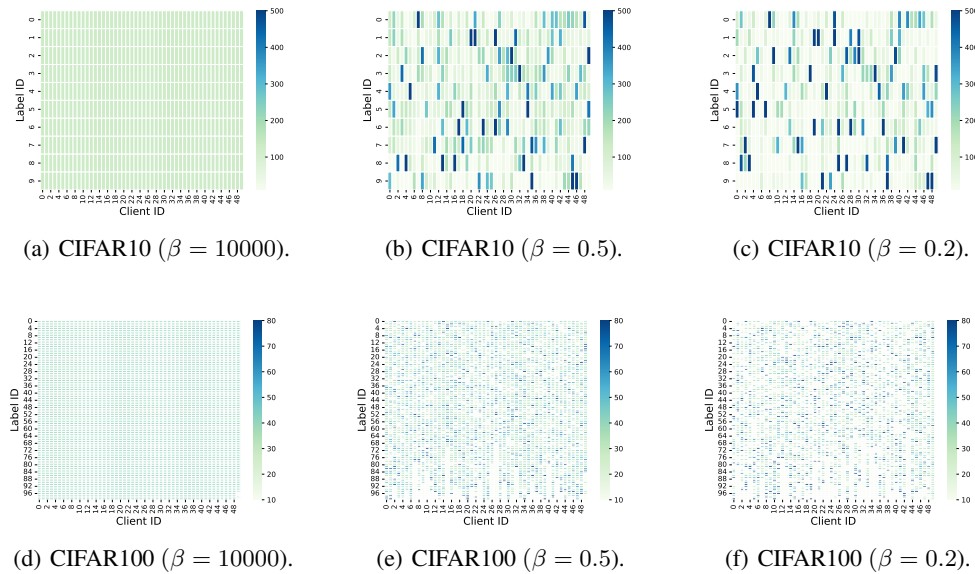

(a) CIFAR10 ($\beta = 10000$).    (b) CIFAR10 ($\beta = 0.5$).    (c) CIFAR10 ($\beta = 0.2$).

(d) CIFAR100 ($\beta = 10000$).    (e) CIFAR100 ($\beta = 0.5$).    (f) CIFAR100 ($\beta = 0.2$).

Figure 7: Heatmap of data distribution under Dirichlet distribution with different $\beta$. The empty color denotes there is no sample of a category in a client, indicating the PCDD situation.

Table 8: Mean and std of averaged personal and generic performance on all settings on the four datasets of FedAvg, best baselines, and our FedGELA. we run three different seeds and calculate the mean and std for all methods.

| Method | SVHN | | CIFAR10 | | CIFAR100 | | Fed-ISIC2019 | |
|---|---|---|---|---|---|---|---|---|
| # Metric | PA | GA | PA | GA | PA | GA | PA | GA |
| FedAvg | $94.09_{\pm0.15}$ | $87.39_{\pm0.20}$ | $77.51_{\pm0.29}$ | $62.04_{\pm0.26}$ | $62.78_{\pm0.43}$ | $58.54_{\pm0.39}$ | $77.27_{\pm0.19}$ | $73.59_{\pm0.17}$ |
| Best Baseline | $95.18_{\pm0.19}$ | $88.85_{\pm0.21}$ | $80.61_{\pm0.33}$ | $66.07_{\pm0.24}$ | $64.28_{\pm0.46}$ | $60.31_{\pm0.32}$ | $78.91_{\pm0.13}$ | $74.98_{\pm0.21}$ |
| FedGELA (ours) | $96.12_{\pm0.13}$ | $90.61_{\pm0.19}$ | $82.28_{\pm0.16}$ | $67.34_{\pm0.15}$ | $70.28_{\pm0.36}$ | $62.43_{\pm0.28}$ | $79.29_{\pm0.19}$ | $75.85_{\pm0.16}$ |

split the 6 clients into 20 clients and in each round randomly select 10 clients to join the federated training.

### C.3 Training and Algorithm-Specific Params

Since the aim of our work is not to acquire the best performance on the four datasets, we use stable and almost the best training parameters in FedAvg and applied on all other methods. We verify and use SGD as the optimizer with learning rate lr=0.01, weight decay 1e-4, and momentum 0.9. The batch size is set to 100 and the local epoch is 10. We have verified that such learning rates and local epochs are much more stable and almost the best. Note that, only training params are equal with FedAvg, but method-specific parameters like proximal terms in FedProx and contrastive loss in MOON are carefully tuned.

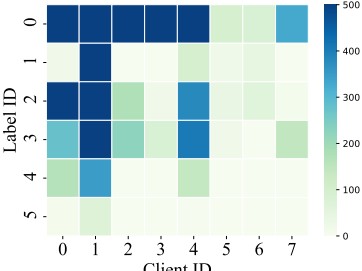

Figure 8: Data distribution of ISIC2019 dataset. The empty color denotes there is no sample of a category in a client, indicating the PCDD situation.

### C.4 Mean and STD

In all our experiments, we run three different seeds and calculate the mean and std for all methods. In Table 3 and Figure 4, we report the both mean and std of results while for other experiments, due to

Table 9: Communication efficiency of FedGELA compared with FedAvg and a range of state-of-the-art methods on CIFAR10 under different settings. Communication efficiency is defined as the communication rounds that need to reach the best global accuracy of FedAvg within curtain rounds. We use '-' to denote the situation that the algorithm can not reach the best accuracy of FedAvg during limited communication rounds.

| Method | IID, Full | | $\beta = 0.5$, Full | | $\beta = 0.1$, Full | | IID, Partial | | $\beta = 0.5$, Partial | | $\beta = 0.2$, Partial | |
|---|---|---|---|---|---|---|---|---|---|---|---|---|
| (CIFAR10) | Commu. | Speedup | Commu. | Speedup | Commu. | Speedup | Commu. | Speedup | Commu. | Speedup | Commu. | Speedup |
| FedAvg | 100 | 1× | 100 | 1× | 100 | 1× | 200 | 1× | 200 | 1× | 200 | 1× |
| FedProx | 42 | 2.38× | 86 | 1.16× | 83 | 1.20× | 105 | 1.90× | 139 | 1.44× | 152 | 1.32× |
| MOON | 42 | 2.38× | 53 | 1.89× | 79 | 1.27× | 98 | 2.04× | 136 | 1.47× | 145 | 1.38× |
| FedRS | 39 | 2.56× | 65 | 1.54× | 84 | 1.19× | 103 | 1.94× | 136 | 1.47× | 126 | 1.59× |
| FedLC | 47 | 2.13× | 45 | 2.22× | 82 | 1.22× | 113 | 1.77× | 118 | 1.69× | 121 | 1.65× |
| FedRep | – | – | – | – | – | – | – | – | – | – | – | – |
| FedProto | – | – | – | – | – | – | – | – | – | – | – | – |
| FedBABU | 63 | 1.59× | 60 | 1.67× | – | – | – | – | – | – | – | – |
| FedRod | 55 | 1.82× | 51 | 1.96× | 77 | 1.30× | 80 | 2.50× | 112 | 1.79× | 142 | 1.41× |
| **FedGELA** | **42** | **2.38×** | **52** | **1.92×** | **67** | **1.49×** | **80** | **2.50×** | **114** | **1.75×** | **119** | **1.68×** |

the limited space, we only report mean results. Therefore in this part, we additionally provide the mean with std of averaged performance on all partitions of FedAvg, best baselines, and our FedGELA in Table 8.

# D  More Information of FedGELA

## D.1  Work Flow of FedGELA

Except for the algorithm of our FedGELA shown in Algorithm 1, we also provide the workflow of FedGELA. As shown in Figure 9, the FedGELA can be divided into three stages, namely the initializing stage, the training stage, and the inference stage. In the initializing stage, the server randomly generates a simplex ETF as the classifier and sends it to all clients. In the meanwhile, clients adjust it based on the local distribution as Eq (4). At the training stage, local clients receive global backbones and train with adapted ETF in parallel. After $E$ epochs, all clients submit personal backbones to the server. In the server, personal backbones are received and aggregated to a generic backbone, which is distributed to all clients in the next round. At the inference stage, on the client side, we obtain a generic backbone with standard ETF to handle the global data while on the client side, a personal backbone with adapted ETF to handle the personal data.

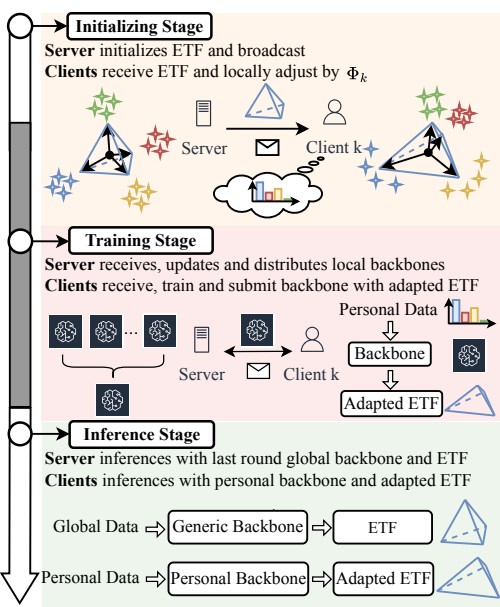

Figure 9: Total framework of FedGELA

## D.2  Communication Efficiency

Communication cost is a much-watched concern in federated learning. Since our algorithm does not introduce additional communication overhead, we compare the number of communication rounds required for all algorithms to reach FedAvg's best accuracy. Since PA is hard to track and highly related to GA as shown in Theorem 3, here we only compare the communication rounds that are required to reach the best GA of FedAvg. As shown in Table 9, we provide communication rounds and speedup to FedAvg compared with a range of the state of the art methods. It can see that P-FL

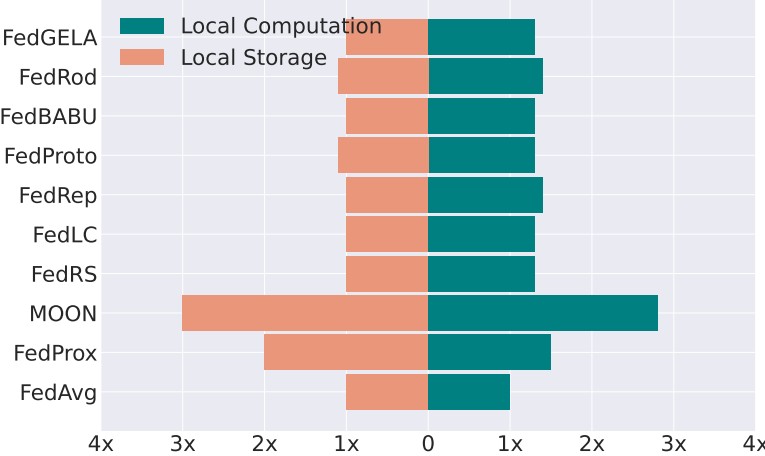

Figure 10: Local computation and storage of FedGELA compared with FedAvg and a range of the state-of-the-art methods.

Table 10: Averaged performance of FedGELA compared with FedAvg and a range of state-of-the-art methods on SVHN under all settings with different backbones, namely Simple-CNN, ResNet18, and ResNet50.

| Method | Simple-CNN | | Resnet18 | | Resnet50 | |
|---|---|---|---|---|---|---|
| #Metric | PA | GA | PA | GA | PA | GA |
| FedAvg | 93.22 | 86.99 | 94.09 | 87.39 | 94.28 | 88.21 |
| Best Baseline | 94.51 | 88.36 | 95.18 | 88.85 | 95.52 | 89.05 |
| FedGELA (ours) | 96.07 | 90.03 | 96.12 | 90.61 | 96.88 | 91.23 |

algorithms are hard to reach the global accuracy of FedAvg since they limit the generic ability of the local model while our FedGELA achieves almost the best communication efficiency in all settings.

### D.3   Local Burden: Storing and Computation

In real-world federated applications, local clients might be mobile phones or other small devices. Thus, the burden of local training can be the bottleneck for clients. In Figure 10, we compute the number of parameters that need to be saved in local clients and the average local computation time in each round. As can be seen, MOON requires triple storing memory than FedAvg, while FedGELA keeps the same level as FedAvg. In terms of local computation time, FedGELA introduces negligible computing time to local training, indicating the efficiency of our method on the local burden concerns.

### D.4   Other Backbones

For SVHN, CIFAR10, and CIFAR100, we conduct all experiments based on ResNet18 (modified by 32x32 input) [8, 17, 33]. Here we adopt more backbones including Simple-CNN and ResNet50 [8, 17, 18] to verify the robustness of our method on different model structures. The Simple-CNN backbone has two 5x5 convolution layers followed by 2x2 max pooling (the first with 6 channels and the second with 16 channels) and two fully connected layers with ReLU activation (the first with 120 units and the second with 84 units. As shown in Table 10, we provide results of FedAvg, best baselines, and our FedGELA on SVHN. As can be seen, with the model capacity increasing from Simple-CNN to ResNet50, the performance is slightly higher. Besides, no matter whether adopting any of the three backbones, our method FedGELA outperforms FedAvg and the best baselines.

Table 11: Performance of FedGELA compared with FedAvg and a range of state-of-the-art methods on two additional real-world challenges, namely FEMNIST and SHAKESPEARE.

| Dataset | FedAvg | | Best Baseline | | FedGELA (ours) | |
|---|---|---|---|---|---|---|
| #Metric | PA | GA | PA | GA | PA | GA |
| FEMNIST | 67.02 | 59.54 | 69.54 | 61.22 | 71.84 | 62.08 |
| SHAKESPEARE | 49.56 | 44.53 | 51.66 | 47.29 | 53.63 | 48.39 |

### D.5 Performance on more real-world datasets

Except for Fed-ISIC2019 used in the main paper, we here additionally test FedGELA on two real-world federated datasets FEMNIST [5] and SHAKESPEARE [28] (two datasets also satisfy the PCDD setting) compared with all related approaches in the paper. FEMNIST includes complex 62-class handwriting images from 3500 clients and SHAKESPEARE is a next-word prediction task with 1129 clients. Most of the clients only have a subset of class samples. With help of LEAF [2], we choose 50 clients of each dataset into federation and in each round we randomly select 10 clients into training. The total round is set to 20 and the model structure is a simple CNN for FEMNIST and a 2-layer LSTM for SHAKESPEARE, respectively. It can be seen in the Table 11, our method achieves best results of both personal and generic performance on the two real-world challenges.

### D.6 Limitations

The design of our method is focused on constraining the classifier in the global server and in the local client by fixing the global classifier as a simplex ETF and locally adapting it to suit personal distribution, which means our method is proposed for federated classification tasks. But the spirit that treating each class or instance equally in global tasks while adapting to personal tasks the local can be applied to more than federated classification tasks. Fixing the classifier as a simple ETF might reduce the norm of stochastic gradients $G$ and benefit global convergence as introduced in Theorem 1 and Theorem 3. However, the limitation is that adapting the local classifier from ETF ($\mathbf{W}^L$) to adapted ETF ($\mathbf{\Phi}_k \mathbf{W}^L$) will introduce additional cost $\|\mathbf{\Phi}_k \mathbf{W}^L - \mathbf{W}^L\|$ as illustrated in the Theorem 1, 2, 3, 4. To verify the influence of the cost and the effectiveness of our method on utilizing waste spaces and mitigating angle collapse of local classifier vectors, we conduct a range of experiments and record performance on both personal and generic and the corresponding angles.