# OpenReview forum: "Federated Learning with Bilateral Curation for Partially Class-Disjoint Data"
_NeurIPS.cc/2023/Conference — NeurIPS 2023 poster_

### Official Review · Reviewer_J5UX · 2023-07-01

**Soundness:** 3 good
**Presentation:** 3 good
**Contribution:** 3 good
**Rating:** 6
**Confidence:** 2

**Summary:**

This paper addresses a challenge in Federated Learning referred to as partially class-disjoint data (PCDD), where each client contributes a part of classes (instead of all classes) of samples. Without full classes, the local objective will contradict the global objective, yielding the angle collapse problem for locally missing classes and the space waste problem for locally existing classes. This is a real-world challenge since it is not uncommon for example that some classes will be well sampled in certain regions, but not others. Prior art mainly focus on the general heterogeneity without specially considering partially class disjoint challenges. Without full classes, the local objective will contradict the global objective, yielding the angle collapse for locally missing classes and the waste of space for locally existing classes. The goal is to achieve holistic improvement in the bilateral views (both global view and local view) of federated learning.

The authors propose FedGELA where the classifier is globally fixed as a simplex ETF while locally adapted to the personal distributions. Globally, FedGELA provides fair and equal discrimination for all classes and avoids inaccurate updates of the classifier, while locally it utilizes the space of locally missing classes for locally existing classes.

The proposed approach builds upon simplex equiangular tight frame (ETF), which provides each class the same classification angle and generalizes well on imbalanced data. Specifically, in their FedGELA approach, the classifier is globally fixed as a simplex ETF while locally adapted based on the local distribution matrix to utilize the wasted space for the existing classes. In the global view, FedGELA merges class features and their corresponding classifier vectors, which converge to ETF. In the local view. it provides existing major classes with larger feature spaces and encourages to utilize the spaces wasted by locally missing classes.

Contributions are summarized as :
- Study algorithmic implication of a real-world  challenge (partially class-disjoint data (PCDD), namely angle collapse and space waste
- Propose FedGELA and theoretically show the local and global convergence analysis for PCDD with the experimental verification
- Evaluate on multiple benchmark datasets under the PCDD case and a real-world dataset to demonstrate the bilateral advantages of FedGELA over the state of the art methods.



**Strengths:**

1. Related work well covers comparison among a range of FL methods and why PCDD not covered. Examples of why prior art does not address PCDD include: generic federated leaning adopt a uniform treatment of all classes, then attempt mitigate personal differences; personalized federated learning places less emphasis on locally missing classes and selectively shares parameters/prototypes to minimize the impact of personal characteristics. While these methods might directly or indirectly help mitigate the data shifts caused by PCDD, neither achieve holistic improvement for global and local views
2. Performance evaluaton on 3 relevant datasets (SVHN, CIFAR10, CIFAR100), against all the top state of the art algorithms as baselines, and showng it outperforms. FedGELA consistently exceeds all baselines.



**Weaknesses:**

Overall, the paper represents is a solid contribution - well defined problem not addressed by prior art and representative of real-world problem for fedrated learing. Solid treatment of prior art, and differentiation from prior methods.
Weaknesses:
1. Only 1 real PCDD federated application Fed-ISIC2019 was evaluated - however I am not aware of other benchmarks I would recommend.
2. Performance improvements against to the best baseline for all tests were all <3% performance improvement.

**Questions:**

1. Given the performance improvements were in the <3% against the best baselines - can you provide more information? For example does it perform better worse under certain conditions? if so, pls distinguish, and if possible explain what this may imply about either the algorithm and limitations, or perhaps a limitation in the benchmark wrt fully characterizing the real world challenge?

**Limitations:**

No negative societal effects. Limitations explained in Weaknesses section.

---

> ### Author Rebuttal · Authors · 2023-08-09
>
>
> **We really appreciate your positive support and the constructive comments. In the following, we provide the detailed response and hope that can address your concerns. Let W, Q and A denote the shorthand of Weaknesses, Question and Answer respectively.**
>
> > **W1:** Only 1 real PCDD federated application Fed-ISIC2019 was evaluated - however I am not aware of other benchmarks I would recommend.
>
> **A:** To address the reviewer's concern, we **test FedGELA on FEMNIST[1] and SHAKESPEARE[2] (two datasets also satisfy the PCDD setting)** compared with all related approches in the paper. FEMNIST includes complex 62-class handwriting images from 3500 clients and SHAKESPEARE is a next word prediction task with 1129 clients. Most of the clients only have a subset of class samples. With help of LEAF[3], we choose 50 clients of each dataset into federation and in each round we randomly select 10 clients into training. The total round is set to 20 and the model structure is a simple CNN and 2 layer of LSTM for FEMNIST and SHAKESPEARE, respectively. It can be seen **in the following table**, **our method achieves best results of both personal and generic performance on all three real-world challenges**.
>
> | Dataset | split | FedAvg | Best Baseline | FedGELA |
> | --- | --- | --- | --- | --- |
> | SHAKESPEARE | PA | 49.56 **+4.07**  | 51.66 **+1.97** | **53.63** |
> |  | GA | 44.53 **+3.86** | 47.29 **+1.10** | **48.39** |
> | FEMNIST | PA | 67.02 **+4.82** | 69.54 **+2.3** | **71.84** |
> |  | GA | 59.54 **+2.54** | 61.22 **+0.86** | **62.08** |
> | FedISIC-2019 | PA | 77.27 **+2.00** | 78.91 **+0.36** | **79.27** |
> |  | GA | 73.59 **+2.26** | 74.98 **+0.96** | **75.85** |
> |
>
> [1]EMNIST: Extending MNIST to handwritten letters
> [2]William Shakespeare: the complete works
> [3]Leaf: A benchmark for federated settings
>
> > **W2 and Q1:** Performance improvements against to the best baseline for all tests were all <3% performance improvement. Given the performance improvements were in the <3% against the best baselines - can you provide more information? For example does it perform better worse under certain conditions? if so, pls distinguish, and if possible explain what this may imply about either the algorithm and limitations, or perhaps a limitation in the benchmark wrt fully characterizing the real world challenge?
>
> **A:** Thank you for the suggestion. We explain this in the following two points.
>
> 1)**Significant Improvement on Pure PCDD settings.** For the consideration of practical data distribution and cohesion with most previous works, we use dirichlet distribution to split the dataset, which will generate heterogeneity data more than PCDD and might limit the potential better improvement. To further address the reviewer's concern, **we decouple the PCDD setting and the ordinary heterogeneity(Non-PCDD), and conduct the corresponding experiments on pure PCDD situations.** In the following table, we use PxCy to denote the dataset is divided in to x clients with y classes of samples, and in each round, 10 clients are selected into federated training. The training round is 100. According to the experimental results, we can see that our FedGELA achieves **significant improvement especially 18.56% to FedAvg and 10.04% to the best baseline** on CIFAR10(P50C2).
>
> | Dataset(Split) | Metric | FedAvg | Best Baseline | FedGELA |
> | --- | --- | --- | --- | --- |
> | CIFAR10(P10C2) | PA | 92.08 **+3.76** | 94.07 **+1.77** | **95.84** |
> |  | GA | 47.26 **+12.34** | 52.02 **+7.58** | **59.60** |
> | CIFAR10(P50C2) | PA | 91.74 **+3.68** | 93.22 **+2.20** | **95.42** |
> |  | GA | 36.22 **+18.56** | 44.74 **+10.04** | **54.78** |
> | SVHN(P10C2) | PA | 95.64 **+3.11** | 97.02 **+1.73** | **98.75** |
> |  | GA | 69.34 **+14.22** | 76.06 **+7.5** | **83.56** |
> | SVHN(P50C2) | PA | 94.87 **+3.5** | 96.88 **+1.49** | **98.37** |
> |  | GA | 66.94 **+10.24** | 72.97 **+4.21** | **77.18** |
> |
>
> 2)Regarding the potential limitations, we can think that curating the structure of the last layer really builds on top of the power deep neural networks, whose fitting ability is sufficiently powerful. Thus regularzing structure of the classification layer does not hurt the overall model capacity too much, and promote the training calibration under PCDD. But when it comes to the shalow models, regularizing the structure of the classification layer might be too harsh and might hurt the training. Another point that is worth exploring in the future is that in the self-supervised learning, it might not contain the label information avaliable and how to curation the model under the PCDD scenario to promote the training remains unknown.
>
> **We appreciate the reviewer's advice and will include the discussion of this question into the submission for better clarity and future explorations.**

---

> > ### Comment · Reviewer_J5UX · 2023-08-15
> >
> > I have read the authors rebuttal and rebuttals to some of the other reviewers.  The authors' rebuttal addresses well the points raised in my review, and are recommended for inclusion in the final submission, if accepted. I do not have further questions for the authors.

---

> > > ### Author Response · Authors · 2023-08-15
> > >
> > >
> > > We sincerely appreciate the positive support of the reviewer. We will carefully follow your constructive comments and include the corresponding contents in the revision to improve the submission.
> > >
> > > Best,
> > >
> > > The Author of Submission7489

---

### Official Review · Reviewer_SUwJ · 2023-07-04

**Soundness:** 3 good
**Presentation:** 3 good
**Contribution:** 2 fair
**Rating:** 6
**Confidence:** 3

**Summary:**

This paper mainly focuses on the partially class-disjoint data (PCDD) problem in federated learning (FL) settings, which is a common yet challenging problem in distributed data sources. Inspired by a classifier structure (simplex equiangular tight frame, ETF), the authors of the paper propose FedGELA to tackle the PCDD problem. FedGELA is a variant of FedAvg with local model adaptation (personalization): They first define the classifier $W$, which is the ETF that the classifier should converge to. Here, they also take the client local data distribution ($\phi$) into account. Afterwards, the feature extractor $H$ will be optimized locally at each client and communicated via server-client communication. Finally, the global feature extractor and the $W$ at central server, as well as the local feature extractors and the adapted $W$ at clients will be returned.

**Strengths:**

The proposed method is motivated very well. The schematic illustration is also clear. The theoretical analysis is sound. Experiments and the results are good.

**Weaknesses:**

From my understanding, FedGELA focuses only on the alignment in the feature embedding space, which has been done by many previous works (FedGen [1], FedProto [2], …). Therefore this paper lacks significance to some extent. Also, the definition of $W$ (ETF, the global classifier), as well as the locally adapted ones looks straightforward.

**Questions:**

1.	You claim that FedGELA could mitigate PCDD from both Global and Local view. But in Algorithm 1, the global model $H^T$ is simply an average of the client model and there is no server-side optimization. Could you please explain this?
2.	In Table 1, you claim that FedGELA could mitigate the model skew, could you please explain this point in more details? Also, in terms of “save space”, since you are transmitting the whole feature backbone $H$, which is the resnet18 in your experiments, what is the reason of space saving?
3.	Why do you use $H$ to represents 2 different terms, features (Line 114) and global backbone (in the Algorithm)? It’s a bit confusing during reading.
4.	Could you shortly explain the converged angles in your figures? Is there a specific meaning of the values? E.g. in Figure 2.
5.	In Equation 4, you model the client data skew in the label space via $\phi$, which is based on the number of samples from different classes. Have you experimented with other options? This looks a bit too straightforward to me.
6.	FedGen[1] is a method which augment the feature embedding space using a shared feature generator, which could possibly mitigate the issue of “waste of space” by generating synthetic embeddings from minority classes. Could you provide a comparison with this work?
[1] Zhu, Zhuangdi, Junyuan Hong, and Jiayu Zhou. "Data-free knowledge distillation for heterogeneous federated learning." ICML, 2021.
[2] Tan, Yue, et al. "Fedproto: Federated prototype learning across heterogeneous clients." AAAI 2022.
Minors:
1.	Line 115, $E_W$ and $E_H$ should be introduced at their first appearance.
2.	Line 298, the selection of $E_W$ seems to be dataset specific, is there any default values suggestions?



**Limitations:**

FedGELA is only tested in the client data with label skew. Is it also applicable to the data with feature skew?

---

> ### Author Rebuttal · Authors · 2023-08-09
>
> **We really appreciate your constructive comments. Regarding the questions from the reviewer, we provide detailed response as below, and hope that can address your concern. Let's use Q as a shorthand for Question.**
>
> **Weakness and Q6:**
>
> 1)**Technical Innovation.** We would like to kindly argue that our core contributions lie in identifying the practical yet under-explored PCDD problem and proposing a **bilateral curation** method in principle to combat the challenges. Why we need to **curate the local classifier instead of keeping all classifiers same is not studied under PCDD**, and we give the contraction analysis in Eq. (2) and (3). Besides, such a non-parameteric bilateral curation is not straightforward, as the form in the **local curation is not given in previous study and Eq. (4) must insure that global and local convergence hold simultaneously**(See the reply of Q5). Other forms might not enjoy such a theoretical merit.
>
> 2)**Comparison to FedGen.** **FedGELA focuses on bilateral curation in the parameter space, instead of embedding space.** Although parameter space and embedding space might follow the same spirit, in the embedding space, we need to upload prototypes like FedProto or train generator and generate features in each round like FedGen. But our bilateral ETFs reach consensus in advance and can be guaranteed in principle without the transmission of personal classifiers and burden on local training as shown in Algorithm 1 in Page 5. Note that, the calculation of $\Phi$ is negligible. Empirically, FedProto is included in the paper and here we provide the comparision to FedGen on SVHN in the following table. **We will add FedGen as baseline into the submission**.
>
> ||Partition|Metric|FedAvg|FedGen|FedGELA|
> |---|---|---|---|---|---|
> | Full Parti.(10 clients) |IID|PA|93.01|94.02|94.84|
> |||GA|92.61|93.99|94.66|
> ||$\beta=0.5$|PA|93.95|94.47|96.27|
> |||GA|91.24|92.66|93.66|
> ||$\beta=0.1$|PA|98.10|98.22|98.52|
> |||GA|75.24|76.51|78.88|
> |Partial Parti.(50 clients)|IID|PA|91.44|91.47|94.68|
> |||GA|91.29|91.33|93.59|
> ||$\beta=0.5$|PA|92.70|93.67|95.54|
> |||GA|89.29|91.35|93.29|
> ||$\beta=0.2$|PA|95.31|95.77|96.85|
> |||GA|84.70|87.59|89.58|
> |
>
> **Q1:**
> We kindly point out that the global and local views mean FedGELA can improve the performance of both the global model and the local models. It does not suggest that FedGELA has both client-side and server-side optimization.
>
> **Q2:**
> As described in the caption of Table 1, 1) model skew is the bias of local model and global model to their optimal weights. We mitigate this by the curation on the classification heads; 2) "save space" here means saving locally wasted space instead of the storage size of the model. We will emphasize these points in the caption to avoid misunderstanding.
>
> **Q3:**
> $H$ follows the notation in Layer Peeled Model[1] to better retrospect previous theory about ETF. We specially correct this notation for the global backbone in Line 122. We apologize that we haven't highlighted this point and will improve it to avoid the confusion.
>
> **Q4:**
> In Figure 2, the angle converges to a proper value, meaning that the optimal separation structure is approaching. The value of the converged angle reflects the mean seperation between different classes when the algorithm converges.
>
> **Q5:** The selection of $\Phi_k$ should reflect personal class distribution and satisfy a basic rule for federated learning, wherein the aggregation of local classifiers aligns with the global classifier(line 168 in the paper), thereby ensuring the validity of theoretical analyses from both global and local views. The convergence requirement can be rewrite as $\gamma\sum_k p_kQ_k(\frac{n_{k,c}}{n_k})=1$, where $\gamma$ is the scaling constant and $Q_k(\frac{n_{k,c}}{n_k})$ denotes the potential way to select $\Phi$. It is highly preferable for the selection process to avoid transmitting $Q_k(\frac{n_{k,c}}{n_k})$, which might induce additional privacy risks. There are indeed many ways to select $\Phi_k$. **However, setting $Q_k(\frac{n_{k,c}}{n_k})=\frac{n_{k,c}}{n_k}$  and $\gamma=\frac{1}{C}$ is the only potential way we can find to determine $\gamma$ that both satisfy the convergence and privacy requirement.** Besides, we have also considered employing alternative methods like employing an exponential or power function of the number of samples, which need to share  $Q_k(\frac{n_{k,c}}{n_k})$  but achieve similar performance. The related experiments are shown in the following table. **We will disscuss more and highlight our design intuition in the submission.**
>
> |Partition|Metric|$Q_k(x)=e^{x}$|$Q_k(x)=x^{\frac{1}{2}}$|$Q_k(x)=x$(ours)|
> |---|---|---|---|---|
> |IID|PA|95.12|95.43|94.84|
> ||GA|94.32|93.99|94.66|
> |$\beta=0.5$|PA|96.18|95.56|96.27|
> ||GA|93.28|93.22|93.66|
> |$\beta=0.1$|PA|98.33|98.21|98.52|
> ||GA|78.95|77.18|78.88|
> |
>
> **Q7:**
> We'll explain notations when they first appear.
>
> **Q8:**
> We recommand the default value to 10e3. With larger $E_w$ (even 10e7), FedGELA can also perform better than most of the methods and far better than FedAvg.
>
> **Limitaion:**
> We conduct the additional experiments on the PACS dataset[2], which is commonly used for analyzing feature heterogeneity. FedISIC-2019 used in our paper also includes feature shifts as the images are collected by different hospitals in different situations. As shown below, FedGELA remains applicable and achieves commendable performance even under feature heterogeneity. Our speculation is that, the local classifiers trained on distinct feature domains may exhibit bias. Using the optimal separation structure like FedGELA aids in enhancing performance.
>
> |Dataset|split|FedAvg|Best Baseline|FedGELA|
> |---|---|---|---|---|
> |PACS|PA|97.60|98.99|98.65|
> ||GA|82.30|84.42|85.06|
> |Fed-ISIC2019|PA|77.27|78.91|79.27|
> ||GA|73.59|74.98|75.85|
> |
>
> [1]Exploring deep neural networks via layer-peeled model: Minority collapse in imbalanced training.(NeurIPS21)
>
> [2]Deeper, broader and artier domain generalization.(ICCV17)

---

> > ### Author Response · Authors · 2023-08-15
> > **Invitation to rolling discussion for the possible remaining concerns**
> >
> >
> > Dear Reviewer,
> >
> > We have thoroughly considered your comments and provided detailed response to address your concerns about technical novelty, clarification, more comparison and more verification.
> >
> > We would like to ask you whether you have remaining or more concerns, so that we can try our best to **timely answer you** during this reviewer-author discussion phase, instead of giving some incomplete demonstrations when approaching to the deadline of this phase.
> >
> > Best,
> >
> > The Author of Submission7489

---

> > > ### Author Response · Authors · 2023-08-18
> > >
> > >
> > > Dear Reviewer SUwJ:
> > >
> > > We appreciate your questions and suggestions, which helps us improve the submission. As your rating score is negative, we would like to know that whether our detailed responses have addressed your concerns. If not, we would like to have a further discussion and explanation with your remaining questions. As the deadline is approaching and we have not received your response, we really appreciate that the reviewer can feedback your points on this submission and promote the discussion. Thank you very much.
> > >
> > > Best,
> > >
> > > The authors of submission7489

---

> > ### Comment · Reviewer_SUwJ · 2023-08-18
> >
> > The rebuttal from the authors addresses the concerns in my review, and the additional experiments also indicate the effectiveness of their proposed method.

---

> > > ### Author Response · Authors · 2023-08-18
> > >
> > >
> > > Thank you for your positive support. We will carefully follow your comments to improve the submission in the revision.
> > >
> > > Best,
> > >
> > > The authors of Submission7489

---

> ### Comment · Reviewer_SUwJ · 2023-08-18
> **concerns are addressed**
>
> The rebuttal from the authors addresses the concerns in my review, and the additional experiments also indicate the effectiveness of their proposed method.

---

> > ### Author Response · Authors · 2023-08-18
> >
> >
> > Thank you very much for your confirmation. If the concerns have been addressed, would you like to raise the rating score? We will carefully follow your comments and include all the experiments and discussions in the revision.
> >
> > Best,
> >
> > The authors of Submission7489

---

### Official Review · Reviewer_UNoN · 2023-07-06

**Soundness:** 3 good
**Presentation:** 3 good
**Contribution:** 3 good
**Rating:** 5
**Confidence:** 3

**Summary:**

This paper introduces a novel Federated Learning Algorithm to address the Partially class-disjoint data (PCDD) problem. The approach is based on the simplex equiangular tight frame (ETF) phenomenon to solve the angle collapse issue and introduces a second projection to personalize an adapted structure to save space. The main contributions can be summarized in three aspects: identifying the angle collapse and space waste challenges in the PCDD problem, introducing the novel FedGELA algorithm, and conducting a range of experiments to evaluate its performance. The paper also includes a theoretical analysis with convergence analysis.

**Strengths:**

1. The paper is well-written with a proper structure and clear explanations. The presentation of the authors' ideas is easy to follow due to the effective use of figures and notations.
2. The methodology of the FedGELA algorithm is interesting, and the mathematical deductions are sufficient. The algorithm is clear and provides enough information for reproducibility.
3. The algorithm has been wisely experimented, and the plots are suitable and clear.



**Weaknesses:**

1. The authors claim that "none of the existing methods can intrinsically mitigate PCDD challenges to achieve holistic improvement in the bilateral views of federated learning." However, the PCDD problem seems closely related to the general non-iid (non-independent and identically distributed) problem. The main differences between these two problems have not been explained.
2. Based on my understanding, if PCDD is different from the non-iid problem, it should perhaps be related to the multi-label problem. However, the presentation of the paper, the experimental data, and the methods of experimental comparison all tend to be more inclined towards non-iid problems. Non-iid is a common problem setting, which contradicts the first author's claim of contributions.
3. The performance improvement is limited.
4. I disagree with the statement that "restricting local structure will waste feature space and limit the training of the local model on existing classes." I believe the notion of "waste of space" is unfounded as it appears to have no impact on computational efficiency or performance improvement.

Conclusion:
The methodology and algorithm presented in this paper are interesting, and the paper is written in high quality. However, there seems to be an important flaw in the problem setting. PCDD appears not to be a new issue but rather a non-iid problem under some special conditions.


**Questions:**

see weaknesses

**Limitations:**

see weaknesses

---

> ### Author Rebuttal · Authors · 2023-08-09
>
> **We really appreciate your positive support and the constructive comments. Regarding the weakness mentioned by the reviewer, we provide the detailed response as below, and hope that can address your concern. Let W denotes the shorthand of Weaknesses.**
>
> **Reply to W1 and W2:**
> 1)We would like to explain a bit about the PCDD problem. **It actually belongs to the data heterogeneity case, but does have a very unique characteristic different from the ordinary heterogeneity problem.** That is, if each client only has a subset of classes, it does not share the optimal Bayes classifier with the global model that considers all classes on the server side. While in the ordinary heterogeneity where local clients have all classes of samples but only differ in the class distributions, they do share an optimal Bayes classifier including the global model on the server side.
> 2)Regarding the claim of the first contribution, the contextual description (Line 63) is for the PCDD problem (Line 64). We did not mean to imply that Non-IID is yet under-explored.
> **In the revision, we will carefully consider the reviewer's question and add more explanation in terms of the relationship between PCDD and the data heterogeneity (i.e., multiple Non-IID distributions) for clarity.**
> We will refine this description to avoid misunderstanding.
>
> **Reply to W3:**
> We would like to kindly argue about the effectiveness of our method by the following three points.
>
> 1)We must note that the reported average improvement of 1.5% to the best baseline encompasses all settings and all datasets, including both Non-IID and IID scenarios. Acutally, there is a marginal space for all algorithms to enhance FedAvg in IID situations, while in Non-IID situations, our method has a larger improvement compared to existing approaches, particularly with a 7% and 11.39% generic improvement over FedAvg on CIFAR10, and a 2.12% and 2.58% generic improvement over the best baseline on CIFAR100. These results demonstrate the effectiveness of FedGELA.
>
> 2)For the consideration of practical data distribution and cohesion with most previous works, we use dirichlet distribution to split the dataset, which will generate heterogeneity data more than PCDD and might limit the potential better improvement. **To further address the reviewer's concern, we decouple the PCDD setting and the ordinary data heterogeneity (Non-PCDD), and conduct the corresponding experiments on pure PCDD settings.** In the following table, we use PxCy to denote the dataset is divided in to x clients with y classes of samples, and in each round, 10 clients are selected into federated training. The training round is 100. According to the experimental results, we can see that our **FedGELA achieves significant improvement especially 18.56% to FedAvg and 10.04% to the best baseline on CIFAR10(P50C2)**.
>
> | Dataset(Split) | Metric | FedAvg | Best Baseline | FedGELA |
> | --- | --- | --- | --- | --- |
> | CIFAR10(P10C2) | PA | 92.08 **+3.76** | 94.07 **+1.77** | **95.84** |
> |  | GA | 47.26 **+12.34** | 52.02 **+7.58** | **59.60** |
> | CIFAR10(P50C2) | PA | 91.74 **+3.68** | 93.22 **+2.20** | **95.42** |
> |  | GA | 36.22 **+18.56** | 44.74 **+10.04** | **54.78** |
> | SVHN(P10C2) | PA | 95.64 **+3.11** | 97.02 **+1.73** | **98.75** |
> |  | GA | 69.34 **+14.22** | 76.06 **+7.50** | **83.56** |
> | SVHN(P50C2) | PA | 94.87 **+3.50** | 96.88 **+1.49** | **98.37** |
> |  | GA | 66.94 **+10.24** | 72.97 **+4.21** | **77.18** |
> |
>
> 3)Additionally, our algorithm is **easy to reproduce, requiring almost zero burden** in terms of local storage, local computation, and communication costs compared to FedAvg. Unlike other methods, our approach **does not require fine-tuning for personalization**. This makes it more accessible and practical for implementation in real-world scenarios.
>
> **Reply to W4:**
> Probably, it is because our words "restricting local structure" that confuses the reviewer. Actually this sentence discusses that if we align the global classifier structure as the local classifier structure, it does greatly limit the performance of personalization, since under PCDD, the global model (with the support of full classes) and the local models (with the support of a subset of classes) do not intrinsically share an optimal Bayes classifier (please refer to the explanation for W1 & W2). Recent work FedRod [1] that configures different classifiers on the server and client sides also supports this point.
> Regarding "waste of space", it is a further possible explanation about why aligning the structures under PCDD induces the performance degeneration in terms of the PA metric (for personalization). An intuitive illustration is shown in Figure 2. Empirically as shown in Table 4, aligning the local classifier structure as the global structure, namely ETF (we denote this method FedGE) can not achieve best personal performance. Especially on the real-world PCDD dataset Fed-ISIC2019, the restricting greatly limits the personal performance. Here we show part of results of Table 4:
>
> |  | GE | LA | CIFAR100 |  |  |  | Fed-ISIC2019 |  |
> | --- | --- | --- | --- | --- | --- | --- | --- | --- |
> |  | #Partition |  | Full Parti. |  | Partial Parti. |  | Real World |  |
> |  | #Metric |  | PA | GA | PA | GA | PA | GA |
> | FedAvg | - | - | 69.09 | 62.80 | 56.46 | 54.28 | 77.27 | 73.59 |
> | FedGE | $\checkmark$ | - | 71.46 | 66.02 | 62.67 | 58.98 | 69.88 | 75.54 |
> | FedGELA | $\checkmark$ | $\checkmark$ | 74.23 | 66.05 | 66.33 | 58.81 | 79.27 | 75.85 |
> |
>
> **We will refine this sentence to clarify our meaning and provide the detailed explanations about our statement.**
> [1]On bridging generic and personalized federated learning for image classification

---

> > ### Author Response · Authors · 2023-08-18
> >
> >
> > Dear Reviewer UNoN:
> >
> > As you do have a few arguments about some points in our submission, we would like to kindly ask whether our explanations address your concerns. If not, we would like to have a further discussion with you. We appreciate the reviewer's challenges on some points, which make us improve the submission, and welcome your further discussion. Thank you very much.
> >
> > Best,
> >
> > The author of submission7489

---

### Official Review · Reviewer_nvCV · 2023-07-23

**Soundness:** 3 good
**Presentation:** 3 good
**Contribution:** 2 fair
**Rating:** 6
**Confidence:** 2

**Summary:**

The authors study the problem of federated learning over partially class-disjoint data and propose using equiangular tight frame (ETF) techniques that allows achieving better performance in both the global and personal learning tasks. They show that the existing federated learning approaches suffer either from angle collapse for locally missing classes or from waste of space for locally existing classes, and propose their approach FedGELA which solves both the issues.

**Strengths:**

+ Extensive experiments comparing the proposed approach, FedGELA, with the existing federated learning approaches.
+ Detailed theoretical analysis of the proposed approach.
+ Highlighting the issues of angle collapse and waste-of-space in the federated learning with partially class-disjoint data.

**Weaknesses:**

- Borrows the existing EFT techniques and hence the novelty seems to be limited.
- Overall improvement in average accuracy is marginal (~1.5%) over the existing approaches.

**Questions:**

I like the experimental evaluations and thorough comparison with the prior federated learning works, but I'm concerned about the algorithmic novelty of the proposed approach. Most of the techniques seem to be adapted from the prior known literature on EFT. Can the authors highlight the technical difficulties in directly applying the prior techniques in solving the PCDD problem?

**Limitations:**

I don't think there are any negative societal impacts of this work.

---

> ### Author Rebuttal · Authors · 2023-08-09
>
> **Thanks for your positive support and the constructive comments. Regarding the questions and weaknesses mentioned by the reviewer, we provide the point-to-point response as below, and hope that can address your concern.  Let Q, W and A denote the shorthand of Question, Weaknesses and Answer respectively.**
> > **W1 and Q1:** Borrows the existing ETF techniques and hence the novelty seems to be limited. I like the experimental evaluations and thorough comparison with the prior federated learning works, but I'm concerned about the algorithmic novelty of the proposed approach. Most of the techniques seem to be adapted from the prior known literature on ETF. Can the authors highlight the technical difficulties in directly applying the prior techniques in solving the PCDD problem?
>
> **A:** The main problem in addressing the PCDD dilemma with ETF lies in the inability to remedy the loss in the clients. Although ETF can guarantee the global optimum on the server side, from the local view, only a subset of classes are present, and their optimal Bayes classifier is no longer shared with the server, where all classes are considered. Note that, this is the most distinction for PCDD from the ordinary heterogeneity (Non-PCDD) where all classes appear but only differ in the distribution shift.
> How to deal with this problem is a biggest technical challenge different from directly using ETF. This is the reason why we propose the **Bilateral Curation** in principle. To further show this distinction, we summarize the comparision between our FedGELA and directly using ETF (termed as FedGE for simplicity) in the following table.
>
> |  | GE | LA | CIFAR100 |  |  |  | Fed-ISIC2019 |  |
> | --- | --- | --- | --- | --- | --- | --- | --- | --- |
> |  | #Partition |  | Full Parti. |  | Partial Parti. |  | Real World |  |
> |  | #Metric |  | PA | GA | PA | GA | PA | GA |
> | FedAvg | - | - | 69.09 | 62.80 | 56.46 | 54.28 | 77.27 | 73.59 |
> | FedGE | $\checkmark$ | - | 71.46 | 66.02 | 62.67 | 58.98 | 69.88 | 75.54 |
> | FedGELA | $\checkmark$ | $\checkmark$ | 74.23 | 66.05 | 66.33 | 58.81 | 79.27 | 75.85 |
> |
>
> As can be seen, FedGELA consistently maintains the advantage in the metric of PA (personalization performance for local clients), while FedGE is even significantly worse than FedAvg on the real-world dataset Fed-ISIC2019 in the metric of PA.  Another noticeable point is that we are the first to employ ETF techniques into FL and before us, the corresponding convergence analysis is unknown. We provide both local and global convergence guarantee for naively combining ETF with FedAvg(FedGE) and our FedGELA.
>
> **We will follow the reviewer's advice to further highlight the technical challenge and add these discussion in the submission.**
>
> > **W2:** Overall improvement in average accuracy is marginal (~1.5%) over the existing approaches.
>
> **A:** 1)We must note that the **reported average improvement of 1.5% encompasses all settings, including both IID and Non-IID scenarios**. Acutally, there is a marginal space for all algorithms to enhance FedAvg in IID situations, while in Non-IID situations, our method has a larger improvement of compared to existing approaches, particularly with a 7% and 11.39% generic improvement over FedAvg on CIFAR10, and a 2.12% and 2.58% generic improvement over the best baseline on CIFAR100. These results demonstrate the effectiveness of FedGELA.
> **To further address the reviewer's concern, we decouple the PCDD setting and the ordinary data heterogeneity (Non-PCDD), and conduct the corresponding experiments on pure PCDD settings.** In the following table, we use PxCy to denote the dataset is divided into x clients with y classes of samples, and in each round, 10 clients are selected into federated training. The training round is 100. According to the experimental results, we can see that our **FedGELA achieves significant improvement especially 18.56% to FedAvg and 10.04% to the best baseline on CIFAR10(P50C2).**
>
> | Dataset(Split) | Metric | FedAvg | Best Baseline | FedGELA |
> | --- | --- | --- | --- | --- |
> | CIFAR10(P10C2) | PA | 92.08 **+3.76** | 94.07 **+1.77** | **95.84** |
> |  | GA | 47.26 **+12.34** | 52.02 **+7.58** | **59.60** |
> | CIFAR10(P50C2) | PA | 91.74 **+3.68** | 93.22 **+2.20** | **95.42** |
> |  | GA | 36.22 **+18.56** | 44.74 **+10.04** | **54.78** |
> | SVHN(P10C2) | PA | 95.64 **+3.11** | 97.02 **+1.73** | **98.75** |
> |  | GA | 69.34 **+14.22** | 76.06 **+7.50** | **83.56** |
> | SVHN(P50C2) | PA | 94.87 **+3.50** | 96.88 **+1.49** | **98.37** |
> |  | GA | 66.94 **+10.24** | 72.97 **+4.21** | **77.18** |
> |
>
> 2)Additionally, our algorithm is **easy to reproduce, requiring almost zero burden** in terms of local storage, local computation, and communication costs compared to FedAvg. Unlike other methods, our approach **does not require fine-tuning for personalization**. This makes it more accessible and practical for implementation in real-world scenarios.

---

> > ### Comment · Reviewer_nvCV · 2023-08-16
> >
> > Thank you for providing clarifications. I have no further questions for the authors.

---

> > > ### Author Response · Authors · 2023-08-16
> > >
> > >
> > > Thank you very much for your confirmation. We will carefully include the contents regarding your suggestion in the revision.
> > >
> > > Best,
> > >
> > > The Author of Submission7489

---

### Author Rebuttal · Authors · 2023-08-10

We would like to thank all the reviewers(nvCV, UNoN, SUwJ and J5UX) for their thoughtful suggestions on our paper, and appreciate that the reviewers have multiple positive impressions of our work, including:
- **well defined problem (J5UX) and a clear motivation (SUwJ and J5UX)**
- **a novel and interesting algorithm (UNoN)**
- **solid theoretical justification (nvCV and UNoN) and sufficient mathematical deductions (SUwJ)**
- **extensive and reasonable experiments(nvCV, UNoN and J5UX) with good results (SUwJ)**
- **well-written paper with a proper structure (UNoN) and clear illustration (SUwJ).**

We provide a summary of our responses, and we will add all corresponding discussions, reviewer-recommended related work and experimental results into the manuscript. For detailed responses, please refer to the feedback of each comment/question point-by-point.

**Introduction and Related Works:**

- We clarify some statements like "global and local view", "model skew" and "save space".(for the question of the Reviewer J5UX)
- We explain the relationship between PCDD and Non-IID and the difference between PCDD and traditional label heterogeneity in federated learning.(for the question of the Reviewer UNoN)
- We further analyze the statements "waste of space" and "converged angles". (for the question of the Reviewers UNoN and SUwJ respectively)

**Method:**

- We highlight and clarify the technique difficulties in applying simplex ETF when solving PCDD challenges. (for the question of the Reviewers nvCV)
- We further analyze the chosen of the personal distribution matrix. (for the question of the Reviewer SUwJ)
- We highlight our technical innovation and provide the comparison to FedGen. (for the question of the Reviewer SUwJ)
- We add more descriptions about some notations. (for the question of the Reviewer SUwJ)

**Experiments:**

- We add FedGEN into baselines. (for the question of the Reviewer SUwJ)
- We decouple the PCDD settings and the ordinary (Non-PCDD) data heterogeneity and verify our algorithm at pure PCDD settings to further show the effectiveness. (for the question of the Reviewers nvCV, UNoN and J5UX)
- We verify our algorithm on two more real-world PCDD federated challenges. (for the question of the Reviewer J5UX)
- We provide suggestions about choosing the hyper-parameter. (for the question of the Reviewer SUwJ)
- We test our method under feature heterogeneity. (for the question of the Reviewer SUwJ)

**We appreciate all reviewers’ time and effort again. We will add all corresponding discussions, reviewer-recommended related work and experimental results into the manuscript. We are looking forward to your reply!**

---

### Decision · Program_Chairs · 2023-09-21

**Decision:**

Accept (poster)

**Comment:**

This paper tackles federated learning over class-disjoint data and proposes a technique based on equiangular tight frame to increase performance on a number of benchmarks. The authors analyze the downsides of existing approaches wrt locally disjoint classes and show that the proposed method improves both. Reviewers found the problem interesting and the method to be well-defined and effective, though some had concerns about the novelty of the method vs. novelty of the application and the size of the performance improvement. While I think this paper is borderline, I lean towards acceptance.